# Assessing the introduction risk of vector-borne animal diseases for the Netherlands using MINTRISK: A Model for INTegrated RISK assessment

Clazien J. de Vos[1]*, Wil H. G. J. Hennen[2], Herman J. W. van Roermund[1], Sofie Dhollander[3], Egil A. J. Fischer[1¤], Aline A. de Koeijer[1]

1 Wageningen Bioveterinary Research, Wageningen University & Research, Lelystad, The Netherlands,
2 Wageningen Economic Research, Wageningen University & Research, Den Haag, The Netherlands,
3 European Food Safety Authority, Parma, Italy

¤ Current address: Faculty of Veterinary Medicine, Utrecht University, TD Utrecht, The Netherlands
* clazien.devos@wur.nl

## Abstract

To evaluate and compare the risk of emerging vector-borne diseases (VBDs), a Model for INTegrated RISK assessment, MINTRISK, was developed to assess the introduction risk of VBDs for new regions in an objective, transparent and repeatable manner. MINTRISK is a web-based calculation tool, that provides semi-quantitative risk scores that can be used for prioritization purposes. Input into MINTRISK is entered by answering questions regarding entry, transmission, establishment, spread, persistence and impact of a selected VBD. Answers can be chosen from qualitative answer categories with accompanying quantitative explanation to ensure consistent answering. The quantitative information is subsequently used as input for the model calculations to estimate the risk for each individual step in the model and for the summarizing output values (rate of introduction; epidemic size; overall risk). The risk assessor can indicate his uncertainty on each answer, and this is accounted for by Monte Carlo simulation. MINTRISK was used to assess the risk of four VBDs (African horse sickness, epizootic haemorrhagic disease, Rift Valley fever, and West Nile fever) for the Netherlands with the aim to prioritise these diseases for preparedness. Results indicated that the overall risk estimate was very high for all evaluated diseases but epizootic haemorrhagic disease. Uncertainty intervals were, however, wide limiting the options for ranking of the diseases. Risk profiles of the VBDs differed. Whereas all diseases were estimated to have a very high economic impact once introduced, the estimated introduction rates differed from low for Rift Valley fever and epizootic haemorrhagic disease to moderate for African horse sickness and very high for West Nile fever. Entry of infected mosquitoes on board of aircraft was deemed the most likely route of introduction for West Nile fever into the Netherlands, followed by entry of infected migratory birds.

**Data Availability Statement:** All relevant data are within the manuscript and its Supporting Information files.

**Funding:** The development of MINTRISK was funded by the Dutch Ministry of Agriculture, Nature and Food Quality (KB-12-009.01-001), Wageningen University & Research (KB-33-001-006-WBVR) and the European Food Safety Authority (NP/EFSA/ALPHA/2016/13-CT01; NP/EFSA/ALPHA/2017/10; PO/ALPHA/2019/06). The case study on vector-borne diseases was funded by the Dutch Ministry of Agriculture, Nature and Food Quality (BO-20-009-026). URL Dutch Ministry of Agriculture, Nature and Food Quality: https://www.rijksoverheid.nl/ministeries/ministerie-van-landbouw-natuur-en-voedselkwaliteit URL Wageningen University & Research: https://www.wur.nl/en.htm URL European Food Safety Authority: https://www.efsa.europa.eu/en The funders had no role in study design, data collection and analysis, decision to publish, or preparation of the manuscript.

**Competing interests:** The authors have declared that no competing interests exist.

# Introduction

International trade, globalization, and changes in demographics, land use, and climate all contribute to the geographical expansion of vector-borne diseases (VBDs), not only threatening public health but also livestock health. In the last decades, the Netherlands experienced two major epidemics of VBDs affecting ruminants resulting in severe economic losses for the Dutch livestock industry, namely bluetongue in 2006–2007 and Schmallenberg in 2011–2012 [1,2]. In recent years, several new zoonotic VBDs were detected in the Netherlands, with first tick-borne encephalitis in 2015 [3,4], then Usutu in 2016 [5], and most recently, West Nile fever in 2020 [6,7]. This, combined with the increased incidence of VBDs such as West Nile fever and bluetongue in other European countries [8–14], has led to growing concern about the threat of VBDs for the Dutch livestock industry bringing about the need for tools to evaluate and compare the risk of emerging VBDs to allow for prioritisation in risk management.

A Framework to assess Emerging VEctor-borne disease Risks (FEVER) was developed that addresses all elements that contribute to the risk of vector-borne animal diseases for newly affected areas, i.e. the probabilities and consequences of entry, establishment, spread and persistence [15,16]. FEVER provides a structured approach ensuring consistency and completeness among VBD risk assessments. However, a tool to evaluate and combine the results of the different elements of such a risk assessment in an objective, transparent and repeatable manner was lacking. Such a tool would make it possible to compare diseases for the risk they pose enabling prioritisation of VBDs, and to target, e.g., surveillance and vaccine development at those diseases that pose the highest threat to the livestock industry or public health.

Available methods to combine the separate elements of a risk assessment into a summarising output parameter range from relatively simple methods such as risk matrices [17] to rather complex methods such as Bayesian belief networks [18]. Where risk matrices do not allow for incorporation of uncertainty, Bayesian belief networks require complex probability matrices that are very data intensive. An intermediate approach is described by Havelaar et al. [19] who used the principles of multi-criteria analysis (MCA) [20] to quantitatively estimate the risk of emerging zoonoses to the Netherlands based on seven input parameters. Since FEVER has many more input parameters, the MCA approach was considered not suitable to arrive at an overall risk estimate in this framework. An alternative approach that also allows for incorporation of uncertainty is the knowledge-based approach (KBA) [21] that was used to summarise the output of pest risk assessments [22–24]. With this approach, expert knowledge on infection biology is used to combine the input parameters using either simple calculations or decision rules, or more complex algorithms that take into account the complexity of the infection processes. KBA is a flexible approach, allowing to adapt the level of detail in the calculations to reflect the different levels of complexity and uncertainty in different modules of the model. For that reason, KBA was used to combine the results of FEVER into an overall risk estimate.

In this paper we describe the resulting calculation tool MINTRISK (Model for INTegrated RISK assessment) and illustrate its application in a risk assessment for emerging VBDs. We used MINTRISK to assess the risk of four VBDs for the Netherlands with the aim to prioritise diseases for preparedness. The outcome of this risk assessment can be used to support policy makers in managing the risk of VBD introduction.

# Material and methods

## MINTRISK

MINTRISK is a semi-quantitative calculation tool based on the FEVER framework [15] to assess the introduction risk of a wide variety of vector-borne livestock diseases that are

transmitted by arthropod vectors. The geographical area for which the introduction risk is assessed is named the area at risk. The regions from where the VBD can be introduced are named the risk regions. The routes along which the VBD can travel from the risk regions to the area at risk are named pathways. The calculations in MINTRISK account for the disease transmission dynamics between vertebrate host animals and arthropod vectors. If a disease is zoonotic, spill over to humans is accounted for only in the impact assessment.

MINTRISK uses six steps to evaluate the introduction risk of vector-borne livestock diseases (Fig 1). These steps include (1) entry, i.e., the rate at which a pathogen is expected to enter the area at risk, (2) transmission, i.e., the ability of the pathogen to spread to susceptible hosts in the area at risk via a competent vector, (3) establishment, i.e., the probability that the pathogen can spread from vector to host and vice versa given the conditions of entry into the area at risk (pathway, time and location), (4) spread, i.e., the extent to which the pathogen is able to spread in the area at risk in a single vector season, (5) persistence, i.e., the likelihood that the pathogen will maintain itself in the area at risk for a prolonged period resulting in endemicity, and (6) impact of the disease on the livestock sector and–if zoonotic–on public health in the area at risk, including economic, socio-ethical, and environmental consequences. Results of MINTRISK are given per step and for three summarizing output parameters, i.e. the

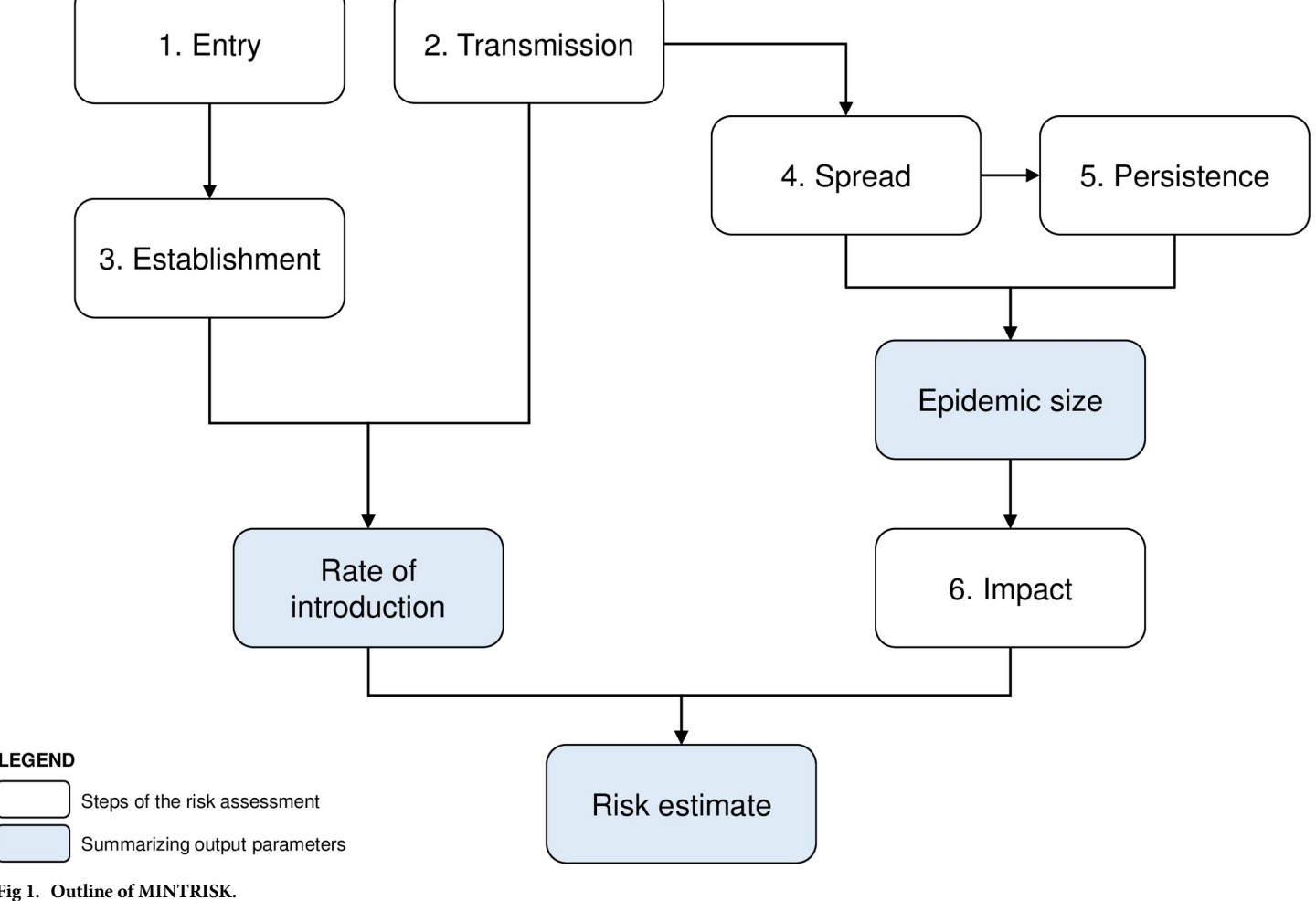

**Fig 1. Outline of MINTRISK.**

rate of introduction, the epidemic size, and an overall risk estimate (Fig 1). The rate of introduction is defined as the expected annual number of entries resulting in successful establishment. The epidemic size returns the estimated number of host animals infected after introduction of the disease, considering a maximum of four vector seasons. The overall risk gives an estimate of the expected economic losses (in Euros) per year considering both the rate of introduction and the economic impact of each individual introduction.

MINTRISK was built as a web-based tool that can be accessed at http://www.wecr.wur.nl/mintrisk2. The model was developed in Microsoft Visual Studio 2017 (.ASP.NET) with C# using DEVexpress components (Visual Studio Dev Essentials) for the user interface. Microsoft SQL Server Management Studio 18 was used to develop the relational database consisting of connected tables in which model parameters and descriptors are defined and where user input and results are stored.

Input into MINTRISK is entered by scoring a set of questions for each step, mostly by choosing from five qualitative answer categories (very low, low, moderate, high, very high) with accompanying quantitative explanation tailored for each question (S1 Appendix). The latter ensures consistent answering of the questions, also when different risk assessors contribute to a comparative risk assessment. The quantitative explanation of the answer categories is mostly on a logarithmic scale, accounting for the prevailing uncertainty when estimating parameter values for infection dynamics of VBDs. In addition, the risk assessor can indicate his/her uncertainty in answering each question, choosing from three categories (low, moderate, high). MINTRISK also offers the opportunity to indicate complete absence of data by offering the option to select 'unknown' when answering the questions.

When performing the model calculations in MINTRISK, the qualitative answers to the questions are converted into numerical values between 0 and 1 using a linear scale. For each answer category, a most likely value has been set with an associated uncertainty interval as indicated in Fig 2. Monte Carlo simulation is used to account for uncertainty in the model calculations with the numerical value for each question being sampled from a triangular distribution representing the uncertainty interval for this question. When the risk assessor has indicated that his/her uncertainty is moderate or high, this uncertainty interval is extended to include the values of one or two adjacent answer categories, respectively. This approach sometimes results in a skewed distribution, especially for the answer categories 'very low' and 'very high', as for those answer categories the uncertainty interval can only be extended on one side. In case the risk assessor has selected the option 'unknown' when answering a question, the numerical value is sampled from a Uniform(0,1) distribution.

The numerical values sampled from the triangular distributions are subsequently log-transformed to obtain quantitative input values for the model calculations (see S1 Appendix). This log-transformation is such that the quantitative input values for the questions correspond with the indicated quantitative explanation of the answer categories selected for the questions. The input parameters are then connected by algorithms that allow for the infection dynamics of VBDs, resulting in a quantitative output value and uncertainty distribution for each step in the model. These algorithms are given in the paragraphs below. The quantitative output of each step is subsequently inverse log-transformed into a semi-quantitative risk score. The inverse log-transformation is such that the resulting risk scores correspond with the most relevant range of quantitative results for each output parameter. This implies that the inverse log-transformation is not by definition on the same scale as the log-transformation at the start of the calculations. A quantitative explanation of the obtained risk scores for each output parameter of MINTRISK is provided in S2 Appendix.

Calculations not only return a semi-quantitative risk score for each individual step in MINTRISK, but also for the summarizing output parameters, i.e. the rate of introduction, the

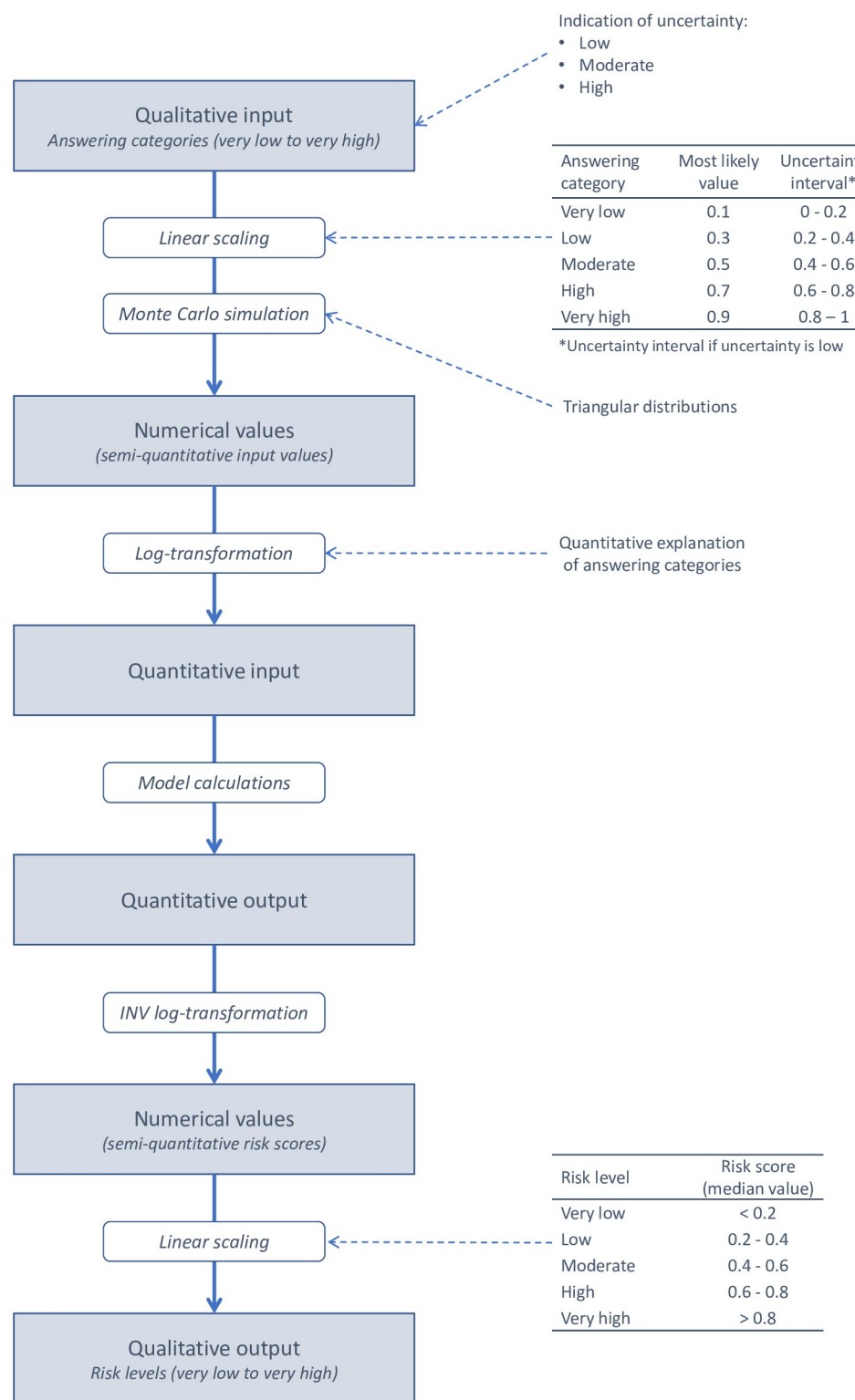

**Fig 2. Schematic overview of how qualitative input into MINTRISK is transformed into quantitative input that can be used for the model calculations and how results are subsequently transformed back into qualitative risk levels.**

epidemic size, and the overall risk estimate (Fig 1). In contrast to the sampled numerical input values, the resulting semi-quantitative risk scores are not bounded by 0 and 1, where a value below 0 indicates a very low to negligible risk and a value above 1 indicates an extremely high risk. Since MINTRISK takes into account uncertainty in input values using Monte Carlo simulation, the results of MINTRISK for the semi-quantitative risk scores are also given by uncertainty distributions. The semi-quantitative risk scores are translated into qualitative risk levels based on the median result (Fig 2).

In the next paragraphs, calculations in MINTRISK are described in detail per step. A comprehensive overview of all parameters in MINTRISK is given in Table 1.

**Entry.** There is usually not one single route along which a pathogen can enter a new area. Therefore, MINTRISK allows the risk assessor to assess the rate of entry via different pathways and from different risk regions. Pathways can either be classified as infected host animals or their products (*host*), infected vectors (*vector*), or–if the infection is zoonotic–infected humans (*human*). The semi-quantitative risk score for the rate of entry is calculated separately for each pathway considering the prevalence in either host animals, vectors, or humans in the risk region, and whether the occurrence of the infection in the risk region is endemic or epidemic. If humans are considered dead-end hosts for the infection, the pathways related to entry of infected humans are ignored by MINTRISK.

The annual rate of entry (*Entry*) is calculated as:

$$Entry = V \times max(P_{end\_pt}, P_{epi\_pt}) \times P_{surv\_transport} \times P_{surv\_PM} \tag{1}$$

where *V* is the annual volume of animals, vectors, humans, or commodities moved along the pathway from the risk region to the area at risk, $P_{end\_pt}$ is the probability of the pathway being infected if the infection is endemic in the risk region whereas $P_{epi\_pt}$ is the probability of the pathway being infected if the infection occurs epidemically, $P_{surv\_transport}$ is the probability that the pathogen will survive in the pathway until arrival in the area at risk, and $P_{surv\_PM}$ is the probability that the pathogen is still present upon arrival in the area at risk despite measures taken to prevent its entry, such as clinical inspection, testing, quarantine, application of acaricides, insecticide spraying, or treatment of products. $P_{end\_pt}$ equals the expected prevalence of infection in the risk region in an endemic situation in either host animals, vectors, or humans, where the value used in the calculations depends on the classification of the pathway (*pt* is *host*, *vector* or *human*). The probability of infection of pathways originating from risk regions with epidemic occurrence of the disease ($P_{epi\_pt}$) is not only based on the expected prevalence in either host animals, vectors, or humans under epidemic conditions, but also on the frequency of epidemics (per year) in the risk region ($F_{epi}$), the fraction of the total area of the risk region that will be affected by an epidemic (*Area*), and the length of the high-risk period in years in the risk region ($HRP_{RR}$), where the high-risk period is defined as the period from first infection in the risk region until detection and notification, during which spread of the infection is not confined by control measures. $P_{epi\_pt}$ is calculated as:

$$P_{epi\_pt} = Min(F_{epi} \times Area \times HRP_{RR} \times Prev_{epi\_pt}, 1) \tag{2}$$

where $Prev_{epi\_pt}$ is the expected prevalence in pathway type *pt* (*animal*, *vector* or *human*).

MINTRISK thus offers the risk assessor the option to indicate whether the VBD is endemic or epidemic in the risk region, where epidemic was defined as the incidental presence of the disease in regions where the disease is normally absent, and endemic was defined as the continuous presence of the disease in the risk region over a longer period (i.e. years). The risk assessor can also decide to enter values for both epidemic and endemic presence. In that case, MINTRISK uses a worst-case approach by selecting the parameter ($P_{end}$ or $P_{epi}$) resulting in

**Table 1. List of parameters used for calculations in MINTRISK.**

| Step | Parameter | Description | Type |
|---|---|---|---|
| **Entry** | | | |
| | $V$ | Annual volume of host animals / vectors / commodities / humans moved along the pathway from the risk region to the area at risk | User input |
| | $P_{surv\_transport}$ | Probability that pathogen will survive in the pathway until arrival in the area at risk | User input |
| | $P_{surv\_PM}$ | Probability that pathogen is still present upon arrival in the area at risk despite preventive measures | User input |
| | $P_{end\_pt}$ | Probability of pathway being infected if disease is endemic in the risk region (equals prevalence of infection in host animals / vectors / humans in endemic situation) | User input |
| | $P_{epi\_pt}$ | Probability of pathway being infected if disease is endemic in the risk region | Calculated |
| | $Prev_{epi\_pt}$ | Prevalence of infection in host animals / vectors / humans in the risk region if disease is epidemic | User input |
| | $F_{epi}$ | Frequency of epidemics (per year) in the risk region | User input |
| | $Area$ | Fraction of the total area of the risk region affected by an epidemic | User input |
| | $HRP_{RR}$ | Length of the high-risk period in years in the risk region | User input |
| | $Entry$ | Annual rate of entry (number of infected entries per pathway) | Calculated |
| **Transmission** | | | |
| | $R$ | Basic reproduction number of the infection in a fully susceptible population with abundant presence of vectors | User input |
| | $F_{susc\_host}$ | Fraction of the host population susceptible to infection in the area at risk | User input |
| | $D_{vector}$ | Reduction factor to account for non-homogeneous distribution of the vector in the area at risk | User input |
| | $R_{opt}$ | Reproduction number of the infection under optimal conditions in the area at risk | Calculated |
| | $RS\_R_{opt}$ | Semi-quantitative risk score for the optimal reproduction number ($R_{opt}$) | Risk score |
| **Establishment** | | | |
| | $P_{inf\_1}$ | Probability of first transmission step (from host to vector or from vector to host) | User input |
| | $P_{inf\_2}$ | Probability of second transmission step (from vector to host or from host to vector) | User input |
| | $Est$ | Probability of establishment in the area at risk | Calculated |
| **Spread** | | | |
| | $R_{eff}$ | Effective reproduction number accounting for the dilution effect | Calculated |
| | $R_{CM}$ | Effective reproduction number when control measures are applied | Calculated |
| | $Dilution$ | Dilution effect due to presence of non-susceptible hosts in the area at risk | User input |
| | $CM_{vector}$ | Effectiveness of vector control measures in reducing spread of the infection | User input |
| | $CM_{host}$ | Effectiveness of other control measures aimed at host animals in reducing spread of the infection | User input |
| | $IG_{season}$ | Number of infection generations in one vector season | User input |
| | $IG_{eff}$ | Effective number of infection generations in one vector season, considering spatial and ecological conditions limiting spread of infection | Calculated |
| | $IG_{det}$ | Number of infection generations until detection of disease | Calculated |
| | $Overlap$ | Overlap between vector abundance and susceptible host animal density | User input |
| | $Local$ | Inhibition of local spread by spatial effects | User input |

*(Continued)*

**Table 1.** (Continued)

| Step | Parameter | Description | Type |
|---|---|---|---|
| | $Mov_{vector}$ | Contribution of vectors to long-distance spread | User input |
| | $Mov_{host}$ | Contribution of host animals to long-distance spread | User input |
| | $HRP_{AaR}$ | Length of the high-risk period in the area at risk (years) | User input |
| | $T_{season}$ | Length of the vector season (fraction of the year) | User input |
| | $PopS$ | Population size of susceptible host animals (epidemiological units) in the area at risk | User input |
| | $Inf_{Total}$ | Total number of infected hosts (epidemiological units) during the first vector season | Calculated |
| **Persistence** | | | |
| | $Inf_{Winter}$ | Number of infected hosts (epidemiological units) of the last infection generation of the vector season (before start of the adverse/winter season) | Calculated |
| | $P_{overwinter}$ | Probability that the infection overwinters until the next vector season per infected host animal present at the end of the vector season | Calculated |
| | $Pers$ | Expected number of infected hosts (epidemiological units) at the start of the next vector season | Calculated |
| **Impact** | | | |
| | $Eco_{DA}$ | Direct agricultural losses per infected epidemiological unit | User input |
| | $Eco_{IA}$ | Indirect agricultural losses per infected epidemiological unit | User input |
| | $Eco_{PH}$ | Economic losses due to human cases per 100 infected epidemiological units (only if zoonotic) | User input |
| | $Eco_{IC}$ | Indirect agricultural losses for the entire area at risk | User input |
| | $Eco_{SE}$ | Economic losses due to side effects for the entire area at risk | User input |
| | $Eco$ | Economic impact (Euros) | Calculated |
| | $RS\_Eco$ | Semi-quantitative risk score for the economic impact ($Eco$) | Risk score |
| | $Soc$ | Socio-ethical impact | Calculated |
| | $Env$ | Environmental impact | Calculated |
| **Summarizing output parameters** | | | |
| | $Intro$ | Annual number of entries resulting in successful establishment in the area at risk | Calculated |
| | $RS\_Intro$ | Semi-quantitative risk score for the rate of introduction ($Intro$) | Risk score |
| | $RS\_Intro_{final}$ | Final semi-quantitative risk score for the rate of introduction | Risk score |
| | $ES$ | Estimated epidemic size (total number of infected epidemiological units over 4 vector seasons) | Calculated |
| | $Risk$ | Overall risk estimate (expected annual economic loss due to introduction of the infection) | Risk score |

the highest semi-quantitative risk score for *Entry* (see Eq 1). This will mostly be $P_{end}$, because its calculation does not include any risk-reducing parameters. If little data is available on the disease situation in the risk region, it is advised to opt for endemic presence in MINTRISK to avoid a false sense of precision.

**Transmission.** The probability of transmission is estimated in an early stage in MIN-TRISK, as there is no need for a full risk assessment if transmission is very low or even negligible. The risk score for the probability of transmission is an indication of the reproduction number *R* under optimal conditions in the area at risk, i.e. in a local area where both vectors and vertebrate host animals are present in sufficient numbers and in a time period in which temperatures favour spread of the infection. This optimal *R* value ($R_{opt}$) is calculated taking

into account the distribution of the vector in the area at risk, i.e. patchy ($<$ 5% of the area) or homogeneous ($\geq$ 5% of the area), and correcting for the protective effect of vaccination or previous exposure to the infection in the host population if applicable. $R_{opt}$ is calculated as:

$$R_{opt} = F_{susc\_host} \times D_{vector} \times R \qquad (3)$$

where $F_{susc\_host}$ is the fraction of the host population that is susceptible to the infection, $D_{vector}$ is the reduction of $R$ if the distribution of the vector is patchy, and $R$ is the basic reproduction number in a fully susceptible host population with abundant presence of vectors. The value of $D_{vector}$ is set to 0.9 if the distribution of the vector in the area at risk is classified as patchy, whereas its value is set to 1 if the vector is homogeneously distributed in the area at risk or if the distribution is unknown. $D_{vector}$ will thus only result in a limited reduction of $R$ if the distribution of the vector is patchy, as transmission in local hotspots can still be efficient. The rate of transmission in larger areas of the area at risk, will however be reduced if vector distribution is patchy.

**Establishment.** The probability of establishment depends on the pathway, local area, and time of entry of the infection in the area at risk and is thus calculated separately for each pathway. For a successful establishment, the infection needs to complete a full transmission cycle, i.e. the infection has to pass from an introduced infected host animal via a local vector to an indigenous host animal, or from an introduced infected vector via an indigenous host animal to a local vector. The probability of establishment (*Est*) is calculated as:

$$Est = P_{inf\_1} \times P_{inf\_2} \qquad (4)$$

where $P_{inf\_1}$ is the probability of the first transmission step occurring, and $P_{inf\_2}$ the probability of the second transmission step occurring. If entry of the pathogen occurs via contaminated animal products, vaccines, or infected humans, the first transmission step needs to consider the most likely route of infection of either local vectors or host animals.

**Rate of introduction.** The rate of introduction, i.e. the expected annual number of entries resulting in successful establishment, is calculated separately for each pathway that is entered into MINTRISK by the risk assessor, using the output parameters of the steps for entry and establishment. The rate of introduction (*Intro*) is calculated as:

$$Intro = Entry \times Est \qquad (5)$$

To define the qualitative risk level for the rate of introduction, also the output of the transmission step is considered, but only when $R_{opt}$ is below 1, because in such situations the infection will fade out even if establishment is successful in first instance. If $R_{opt} < 1$, the qualitative risk level for the rate of introduction is therefore based on the minimum of the semi-quantitative risk scores obtained for $R_{opt}$ and *Intro*, i.e. the minimum of the numerical values for these parameters after inverse log-transformation (Eq 6). More details on the inverse log-transformation are given in S2 Appendix. The final semi-quantitative risk score for the rate of introduction ($RS\_Intro_{final}$) is thus calculated as:

$$RS\_Intro_{final} = \begin{cases} RS\_Intro & \forall\, R_{opt} \geq 1 \\ Min\,(RS\_R_{opt}, RS\_Intro) & \forall\, R_{opt} < 1 \end{cases} \qquad (6)$$

where $RS\_Intro$ is the semi-quantitative risk score for the rate of introduction based on entry and establishment only (after inverse log-transformation of *Intro*, Eq 5), and $RS\_R_{opt}$ is the semi-quantitative risk score for transmission (after inverse log-transformation of $R_{opt}$, Eq 3).

Then, the risk assessor is asked to select a maximum of three pathways to include in the further MINTRISK calculations, which naturally are those pathways that have the highest value for $RS\_Intro_{final}$. All calculations in MINTRISK up till this stage are deterministic, i.e., they are based on the most likely values for each of the entered answers. This is to avoid extensive simulation time in case numerous pathways are entered into the model. However, if one would have extreme uncertainty about a certain pathway, it could be worthwhile to evaluate its impact by including this pathway in one or more test runs, even though it has a relatively low score for $RS\_Intro_{final}$. Pathways not selected at this stage will be ignored in the further model calculations. The final output of the MINTRISK calculations for this section and beyond is based on the selected pathway with the highest output value for $RS\_Intro_{final}$. This pathway can, however, vary between model iterations, because further calculations for the selected pathways do include Monte Carlo simulation to account for uncertainty in input parameters in MINTRISK.

**Spread.** In this step, the extent of spread of the disease in the first vector season is evaluated. The epidemiological unit considered in this step should equal the epidemiological unit for which the basic reproduction number was given in the transmission step (see paragraph on Transmission) and can, e.g., be individual host animals or herds or flocks. The semi-quantitative risk score for spread is based on the total number of epidemiological units that is expected to get infected during the first vector season. This is calculated primarily from the optimal reproduction number ($R_{opt}$) as calculated in the transmission step, the number of infection generations that fit in one vector season ($IG_{season}$), and the number of infection generations until detection of the disease ($IG_{det}$). However, spread can be limited by several aspects which is accounted for in MINTRISK by adjusting the values of $R_{opt}$ and $IG_{season}$. First, the presence of non-susceptible hosts in the area where the infection is present might result in a so-called dilution effect [15,25,26] if infectious vectors also feed on these animals, resulting in an effective reproduction number ($R_{eff}$) that is calculated as:

$$R_{eff} = R_{opt} \times (1 - Dilution) \tag{7}$$

where *Dilution* is the expected dilution effect. The dilution effect is estimated as the proportion of non-susceptible hosts over the total number of hosts bitten by the competent vectors of a given VBD pathogen. This proportion depends on the host preference of the competent vectors and the abundance of different host animal species.

Spatial characteristics can either limit or favour spread of the infection. In MINTRISK, four parameters are considered, i.e. the overlap between vector abundance and host animal density in the area at risk (*Overlap*), inhibition of local spread by spatial effects (*Local*), and contribution of host animals and vectors to long-distance spread ($Mov_{host}$ and $Mov_{vector}$). These parameters tend to affect spread after the first few transmission cycles given successful establishment, whereas they have less influence on the initial transmission of the infection in the area at risk. The rate of transmission is not only determined by the reproduction number, but also by the expected time interval between infection generations. Therefore, these parameters were modelled as factors that affect the number of infection generations in the vector season, which is the reciprocal of the time interval between infection generations. If all four parameters would maximally inhibit transmission, the number of infection generations is reduced by 50%. The effective number of infection generations ($IG_{eff}$) is calculated as:

$$IG_{eff} = IG_{season} \times \frac{3 + Overlap + (1 - Local) + Max(Mov_{vector}, Mov_{host})}{6} \tag{8}$$

Please note that the questions to assess *Local*, $Mov_{vector}$, and $Mov_{host}$ come without accompanying quantitative explanation. The qualitative answers to these questions are converted into numerical values between 0 and 1 using linear scaling and Monte Carlo simulation (Fig 2), after which their values are directly used in the calculations.

As soon as the infection is detected, control measures will be implemented that reduce the transmission of the infection, resulting in a lowered value of the reproduction number ($R_{CM}$) that is calculated as:

$$R_{CM} = \frac{R_{eff}}{CM_{vector} \times CM_{host}} \tag{9}$$

where $CM_{vector}$ is the effect of vector control measures on local and long-distance spread and $CM_{host}$ the effect of other control measures aimed at host animals.

The rate of transmission of the infection is thus expected to differ between the first phase of the epidemic, which is the high-risk period with undetected spread of the infection and no control measures in place yet, and the second phase of the epidemic that starts upon detection of the disease resulting in control measures. In MINTRISK, the length of the first phase is expressed by the number of infection generations until detection of the disease ($IG_{det}$), which is calculated as:

$$IG_{det} = \begin{cases} IG_{eff} & \forall\ HRP_{AaR} \geq T_{season} \\ IG_{eff} \times \left( \dfrac{HRP_{AaR}}{T_{season}} \right) & \forall\ HRP_{AaR} < T_{season} \end{cases} \tag{10}$$

where $HRP_{AaR}$ is the length of the high-risk period in the area at risk expressed in years, and $T_{season}$ is the length of the vector season expressed as a fraction of the year.

In calculating the total number of epidemiological units infected during the first vector season ($Inf_{total}$), MINTRISK takes into account the length of the high-risk period by changing the transmission parameter from $R_{eff}$ to $R_{CM}$ after detection of the disease. This is only relevant if detection is expected to occur in the first vector season, i.e. $HRP_{AaR} < T_{season}$. Furthermore, the tool accounts for the unlikely event that transmission of the infection will result in natural fade out of the disease in small populations by ensuring that the total number of infected epidemiological units will not exceed the population size ($PopS$). $Inf_{total}$ is calculated as:

$$Inf_{total} = Min \left( PopS, \begin{cases} \displaystyle\sum_{i=0}^{IG_{eff}} R_{eff}{}^{i} & \forall\ HRP_{AaR} \geq T_{season} \\ \displaystyle\sum_{i=0}^{IG_{det}} R_{eff}{}^{i} + \left( R_{eff}{}^{IG_{det}} \times \displaystyle\sum_{j=1}^{IG_{eff}-IG_{det}} R_{CM}{}^{j} \right) & \forall\ HRP_{AaR} < T_{season} \end{cases} \right) \tag{11}$$

MINTRISK also calculates the total number of infections of the last infection generation in the vector season ($Inf_{winter}$) as an input for the persistence step (next paragraph). $Inf_{winter}$ is calculated as:

$$Inf_{winter} = Min \left( PopS, \begin{cases} R_{eff}{}^{IG_{eff}} & \forall\ HRP_{AaR} \geq T_{season} \\ R_{eff}{}^{IG_{det}} \times R_{CM}{}^{(IG_{eff}-IG_{det})} & \forall\ HRP_{AaR} < T_{season} \end{cases} \right) \tag{12}$$

**Persistence.** To estimate the likelihood of persistence, the probability that the infection can survive into the next vector season is evaluated. To this end, the probability of overwintering in both the host animal and vector population are addressed, as well as the probability of

overwintering via other mechanisms such as non-zero vector activity [27] or survival in the environment. The risk assessor is asked to score the probability of overwintering via six independent mechanisms: persistent infection in the host, vertical transmission in the host, direct host-to-host transmission, survival of infected adult vectors, vertical transmission in the vector, and overwintering via other mechanisms. The probability that the infection overwinters until the next vector season ($P_{overwinter}$) for each infected host animal that is present at the end of the vector season, is then set equal to the probability of the overwintering route that has the highest risk score. Effectively, this comes down to sorting out the most likely mechanism for overwintering in the area at risk and scoring the probability for this overwintering route. The likelihood of persistence (*Pers*) is then calculated as the expected number of infected epidemiological units at the start of the next vector season:

$$Pers = Inf_{winter} \times P_{overwinter} \tag{13}$$

**Epidemic size.** The expected epidemic size (*ES*) is calculated taking into account both the number of infected epidemiological units in the first vector season ($Inf_{total}$) and the expected number of infected epidemiological units at the start of the next vector season (*Pers*). Different equations are used depending on the length of the high-risk period in the area at risk, because only after detection of the disease, control measures will be put in place resulting in a reduction of the transmission parameter from $R_{eff}$ to $R_{CM}$.

$$ES = Min\left(PopS, \begin{cases} Inf_{total} + Pers \times \left(1 + R_{CM}{}^{IG_{eff}} \times P_{overwinter} + \left(R_{CM}{}^{IG_{eff}}\right)^2 \times P_{overwinter}{}^2\right) \times \sum_{i=0}^{IG_{eff}} R_{CM}{}^i, & \forall\ HRP_{AaR} \leq 1 \\[2ex] Inf_{total} + Pers \times \left(\sum_{i=1}^{IG_{eff}} R_{eff}{}^i + \left(R_{eff}{}^{IG_{eff}} \times P_{overwinter} \times \sum_{i=1}^{IG_{eff}} R_{CM}{}^i\right)\right) \times \left(1 + R_{CM}{}^{IG_{eff}} \times P_{overwinter}\right), & \forall\ 1 < HRP_{AaR} \leq 3 \\[2ex] Inf_{Total} \times \sum_{i=0}^{3} Pers^i, & \forall\ HRP_{AaR} > 3 \end{cases}\right) \tag{14}$$

In calculating the epidemic size, we assumed that the spread of the infection in the first vector season started with a single infected epidemiological unit, and that each infected epidemiological unit at the start of the new vector season will have a similar probability of inducing new infections. If persistence is high, this might result in a large number of infections in the next vector seasons, exceeding the total population size. Therefore, the total population size (*PopS*) was included in Eq 14 to ensure that the epidemic size does not exceed the total population size. The epidemic size is calculated over a total of four vector seasons.

**Impact.** The impact assessment consists of the evaluation of the economic, socio-ethical and environmental consequences related to the introduction and spread of a vector-borne disease. While the economic consequences can be expressed in monetary values, the quantification of socio-ethical and environmental consequences is less straightforward. To avoid the subjective translation of these elements into monetary or utility values, MINTRISK only accounts for the economic consequences in the overall risk estimate. The questions on socio-ethical and environmental consequences were nevertheless included to raise awareness. Results of these sections can be used to indicate the potentially adverse consequences of disease introduction even if economic consequences are limited.

The main variables determining the impact of disease introduction are the number of epidemiological units (host animals and/or farms) infected, the geographical area affected by the disease, the control measures applied to contain or eradicate the pathogen, and–in case the disease is zoonotic–the number of humans infected and the severity of illness. The economic

impact (*Eco*) is calculated as:

$$Eco = \left(Eco_{DA} + Eco_{IA} + \frac{Eco_{PH}}{100}\right) \times ES + Eco_{IC} + Eco_{SE} \tag{15}$$

where $Eco_{DA}$ are the direct agricultural losses due to e.g. morbidity, mortality and production losses and $Eco_{IA}$ are the indirect agricultural losses related to e.g. empty barns and losses in the supplying and delivering industry (e.g. feed companies and slaughterhouses). Both parameters are estimated per epidemiological unit and multiplied with the estimated epidemic size (*ES*) to arrive at the total economic losses at farm level. $Eco_{PH}$ are the economic losses due to human disease related to e.g. medical treatment and reduced economic productivity. These losses only need to be considered if the vector-borne disease is zoonotic. As the number of infected humans is mostly low compared to the number of infected host animals, $Eco_{PH}$ is estimated for the expected number of human cases per 100 infected host animals in order to scale these losses to the epidemic size in host animals. $Eco_{IC}$ are the indirect agricultural economic losses at national or regional level due to presence of the disease. These include costs incurred due to movement standstills, trade restrictions and control measures. $Eco_{SE}$ are economic losses due to side effects, such as reduced tourism in affected areas. $Eco_{IC}$ and $Eco_{SE}$ are assumed to be related to the geographical area affected rather than the number of epidemiological units infected and are therefore both estimated at the national or regional level.

No accompanying quantitative explanation is available when entering the answer categories for the questions in the sections on socio-ethical and environmental impact in MINTRISK. In these sections of MINTRISK, the numerical values sampled are therefore not log-transformed, but directly used to calculate the semi-quantitative risk scores and resulting qualitative risk levels. The socio-ethical impact (*Soc*) is calculated as the maximum risk score entered for each of the five categories of socio-ethical consequences distinguished in MINTRISK, i.e. socio-ethical impact related to the human disease burden, socio-ethical impact related to reduced animal welfare, socio-ethical impact related to disease in pet animals, socio-ethical impact related to culling of livestock to control the disease, and socio-ethical impact related to loss of recreational outdoor space. Likewise, the environmental impact (*Env*) is calculated as the maximum risk score entered for each of the three categories of environmental consequences distinguished in MINTRISK, i.e. environmental impact related to loss of biodiversity, environmental impact related to effects on nature values, and environmental impact related to vector control.

**Overall risk estimate.** The overall risk estimate in MINTRISK provides an indication of the expected annual economic loss due to introduction of the vector-borne disease. This parameter is only calculated at the level of semi-quantitative risk scores. Usually, risk is calculated as the product of probability and impact. Since the risk scores are on a $\log_{10}$ scale, in this model the risk score for the rate of introduction ($RS\_Intro_{final}$) and the risk score for the economic impact ($RS\_Eco$) are summed to obtain the overall risk estimate:

$$Risk = RS\_Intro_{final} + RS\_Eco \tag{16}$$

## Risk assessment

We evaluated the annual introduction risk of four VBDs for the Netherlands. Diseases included were OIE listed [28] and had never occurred in the Netherlands at the time that this assessment was performed. The diseases selected were African horse sickness (AHS), epizootic haemorrhagic disease (EHD), Rift Valley fever (RVF), and West Nile fever (WNF). AHS and EHD were considered because of their close relatedness to bluetongue (BT) [29–31], which caused a huge epidemic in the Netherlands in the period 2006–2008 [9,32]. All three diseases are transmitted by midges (*Culicoides* spp.). The risk assessment of EHD was limited to EHD

**Table 2. Overview of causing pathogens, vertebrate host animals, arthropod vectors, and geographical distribution of four vector-borne diseases[a].**

| Disease | Pathogen[b] | Vertebrate host animal | Vector | Zoonosis | Geographical distribution |
|---|---|---|---|---|---|
| **African horse sickness** | AHS virus (*Orbivirus*, *Reoviridae*) | Equines | *Culicoides* spp. | No | Sub Saharan Africa |
| **Epizootic haemorrhagic disease** | EHD virus serotype 6 (*Orbivirus*, *Reoviridae*) | Deer, bovines, sheep | *Culicoides* spp. | No | Morocco, Algeria, Tunisia, Turkey, Reunion Island, Guadeloupe, Australia, USA |
| **Rift Valley fever** | RVF virus (*Phlebovirus*, *Bunyaviridae*) | Bovines, sheep, goats, wild ungulates, rodents | *Aedes* spp., *Culex* spp., *Anopheles* spp., *Ochlerotatus* spp. | Yes | Africa, Arabian Peninsula |
| **West Nile fever** | West Nile virus (*Flavivirus*, *Flaviviridae*) | Birds, equines | *Culex* spp. | Yes | South, Central and Eastern Europe, Middle East, Asia, Australia, North, Central and South America, Africa |

[a] Sources

African horse sickness [29,31,42,43].

Epizootic haemorrhagic disease [30,33,44,45].

Rift Valley fever [37,46–49].

West Nile fever [8,50–53].

[b] Genus and family of pathogen given between brackets.

virus serotype 6, because of its occurrence in North-Africa and Turkey, where clinical disease in cattle has been reported [33]. WNF was considered, because of the steady expansion of its geographic distribution in Europe and the geographic proximity to the Netherlands [10,13]. RVF is not present in Europe yet and might therefore pose a lower threat to the Netherlands. However, the outbreaks in the Arabian Peninsula in 2000 [34–36] and serological studies indicating the presence of the virus in the Mediterranean (Turkey, North Africa) [37] have raised awareness in Europe for this disease. The long period of silent spread often observed in RVF epidemics [38,39] warrants increased vigilance as unnoticed presence of the virus in the European Union will result in a high introduction risk by trade in live animals. WNF and RVF are both transmitted by mosquito vectors of the genera *Culex*, which are abundantly present in the Netherlands [40,41]. WNF and RVF are both zoonotic diseases. An overview of the diseases, their pathogens, arthropod vectors, susceptible vertebrate hosts, and geographical distribution is given in Table 2.

The risk assessment was an update of an assessment performed in 2015 [16]. We started out with an extensive list of potential pathways for introduction for each of the diseases using the FEVER framework. Only pathways with a non-negligible risk in the qualitative risk assessment were included in MINTRISK (Table 3). Infected humans were not evaluated as pathways for introduction because humans are considered dead-end hosts for both WNF and RVF.

Questions in MINTRISK were answered using information from global databases, scientific literature, and expert opinion. Answers to all questions in the assessment are documented in S3 Appendix. Semi-quantitative risk scores were calculated in MINTRISK using 1,000 iterations.

## Results

### Rate of introduction

The rate of introduction provides an indication of the annual probability of successful entry, i.e. entry of the pathogen resulting in establishment in the area at risk and subsequent spread. The estimated rate of introduction for the pathway contributing most to the introduction risk varied from low for EHD (median risk score 0.32) and RVF (median risk score 0.39) to very

**Table 3. Pathways entered into MINTRISK to assess the introduction risk of each disease.**

| Type | Pathway | AHS | EHD | RVF | WNF |
|---|---|---|---|---|---|
| **Host–animal** | | | | | |
| | Legal trade in livestock/equines | | | X | |
| | Illegal trade in livestock/equines | **X** | **X** | **X** | |
| | Import of zoo animals | X | | | |
| | Movement of competition horses | X | | | |
| | Migratory birds | | | | **X** |
| **Host–product** | | | | | |
| | Biological material including modified live vaccines | X | X | | |
| **Vector** | | | | | |
| | Transport vehicles (aircraft, ship, road transport) | | **X** | **X** | **X** |
| | Containers on aircraft or ship | | | **X** | **X** |
| | Imported products (plant material, tires) | | | X | |
| | Traded animals (livestock, pets) | **X** | **X** | | |
| | Migration of wildlife | | | | |
| | Migratory birds | | | | |

Pathways selected for inclusion in the model calculations are indicated in bold.

high (median risk score 0.87) for WNF (Fig 3). The rate of introduction of AHS was estimated to be moderate (median risk score 0.51). Uncertainty about the rate of introduction was high.

For both AHS and EHD we assumed that at least one of the Palearctic *Culicoides* species in the Netherlands is a competent vector for these viruses and that establishment is thus possible. Infected adult vectors (*Culicoides* spp.) that enter the Netherlands via traded livestock contributed most to the rate of AHS introduction (Fig 3). This was also an important pathway for EHD, together with illegal trade in livestock from Mediterranean countries. Illegal import of livestock contributed most to the rate of introduction of RVF, with infected mosquitoes arriving in the Netherlands in aircraft or containers having a slightly lower risk score. The risk score for these pathways was much higher for WNF (very high and high, respectively), due to the higher numbers of aircraft and containers arriving from infected areas. The rate of introduction via migratory birds was also evaluated as high for WNF.

## Impact

The impact of a VBD is largely dependent on the epidemic size (Fig 4). The estimated epidemic size was very low for AHS (median risk score -0.11); high for RVF (median risk score 0.67); and very high for EHD and WNF (median risk score 0.87 and 1.29, respectively). It should be noted that a negative risk score indicates a very low to negligible impact, whereas a risk score > 1 indicates an extremely high impact (see Material and Methods and S2 Appendix). The epidemic size depends on both the extent of spread in the vector season and the probability of persistence. WNF had the highest risk estimate for both parameters, even though the probability of overwintering of RVF starting from a single infected host was higher for RVF than WNF. However, the probability of persistence was higher for WNF due to a higher expected number of infected animals at the start of the winter season.

The estimated economic impact was very high for all four diseases (Fig 5) with a median risk score of 1 for EHD, 0.93 for RVF, 0.9 for AHS, and 0.87 for WNF. The very high risk scores for AHS, EHD and RVF are mainly explained from the expected trade restrictions (export ban) in case of an outbreak of these diseases, even if only few animals would be

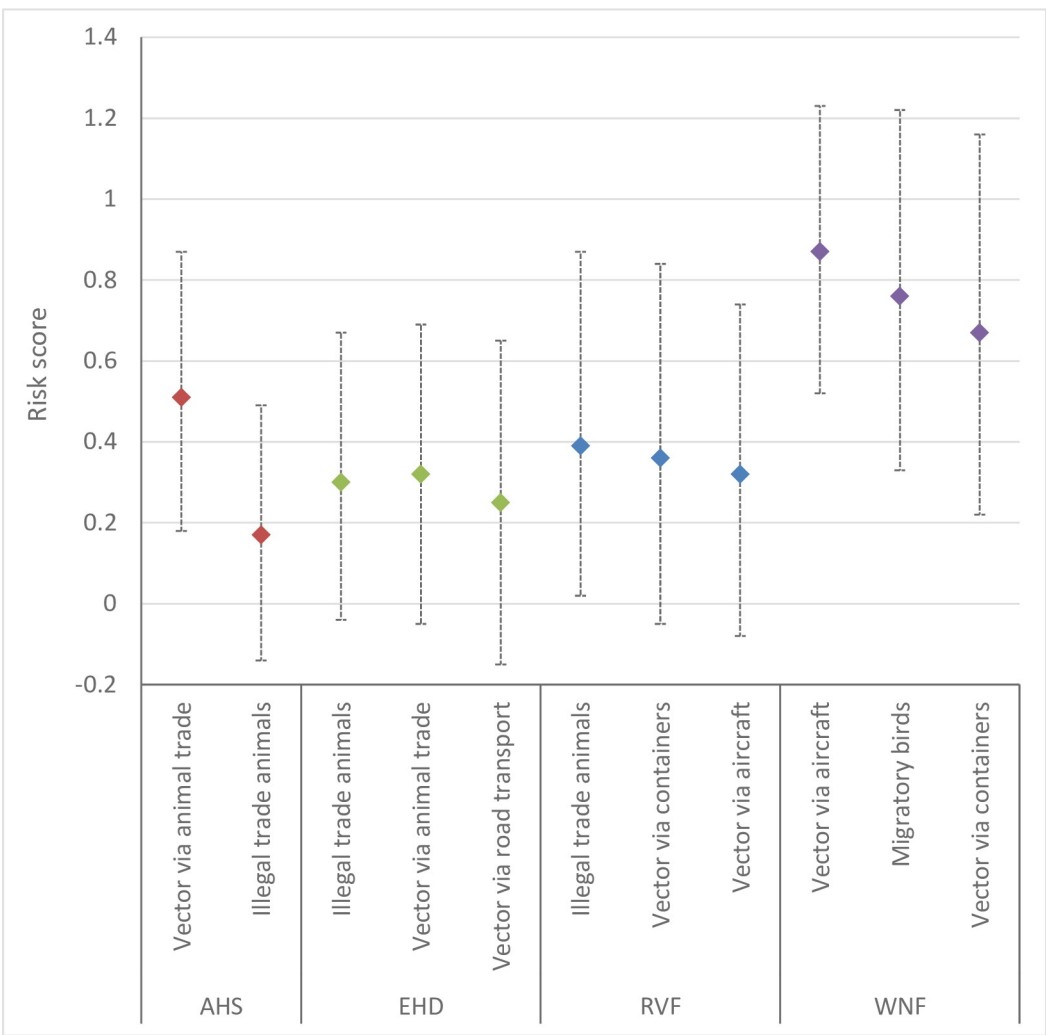

**Fig 3. Risk scores (median values) and their uncertainty (95% uncertainty interval) for the rate of introduction of four vector-borne diseases for the Netherlands via selected pathways.** AHS = African horse sickness; EHD = epizootic haemorrhagic disease; RVF = Rift Valley fever; WNF = West Nile fever.

infected. The estimated epidemic size for AHS was, for example, very low, resulting in limited economic losses due to infected hosts, even though the expected direct agricultural losses per infected host were estimated to be very high. The very high risk score for WNF was unexpected, since all questions related to the economic impact were answered as low or very low. The direct and indirect losses per infected host will be extremely low indeed, since wild birds are the epidemiological units in the MINTRISK calculations for WNF, with equines and humans being spill over hosts only. Furthermore, no trade bans are to be expected if West Nile virus would be detected in the country. The very high economic impact is therefore solely explained from the estimated epidemic size, which was extremely high (Fig 4).

Socio-ethical impact is expected to be high to very high for all diseases except EHD (Fig 5). For WNF and RVF, the high socio-ethical impact is mainly related to societal anxiety due to potentially fatal infections in humans. For AHS, and to a lesser extent for WNF, the high socio-ethical impact can be attributed to the impact on animal welfare, with severe disease in equines also resulting in human suffering given the often-close relationship of humans and

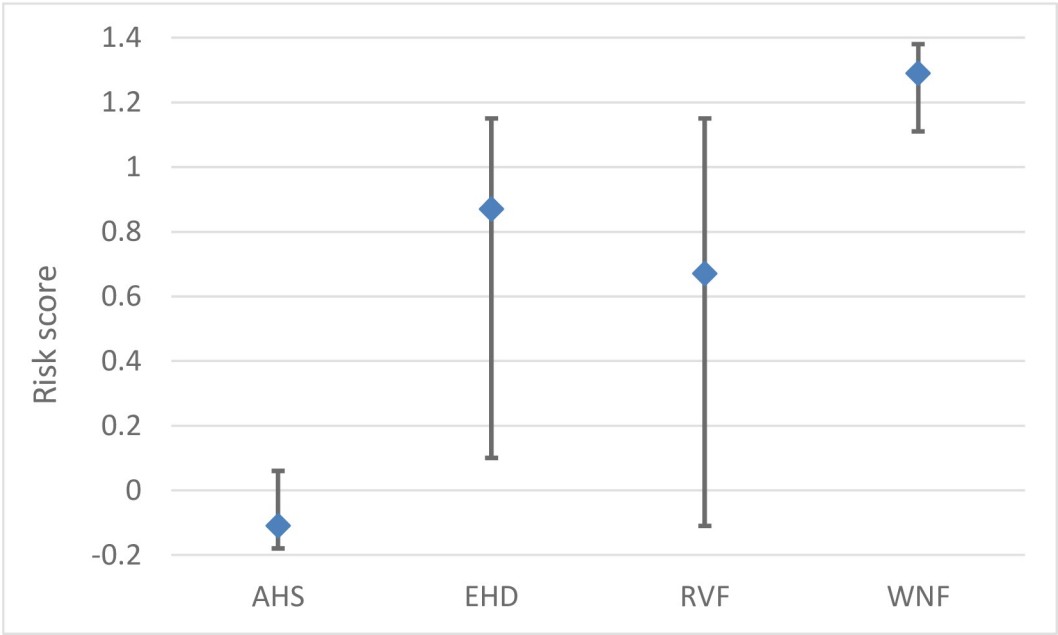

**Fig 4. Risk scores (median values) and their uncertainty (95% uncertainty interval) for the estimated epidemic size of four vector-borne diseases for the Netherlands.** AHS = African horse sickness; EHD = epizootic haemorrhagic disease; RVF = Rift Valley fever; WNF = West Nile fever.

horses in the Netherlands. Furthermore, morbidity and mortality in equines is expected to be very high in case of an AHS outbreak. Moderate environmental impact is expected for EHD and WNF as these diseases are primarily wildlife diseases and might result in severe mortality in vulnerable populations, albeit the estimates include high uncertainty (Fig 5). The environmental impact of AHS and RVF is expected to be very low.

## Overall risk

The overall risk estimate calculated by MINTRISK takes into account the results of all six steps of the FEVER framework. The resulting risk scores can be used to prioritize VBDs for risk management. The overall risk estimate was very high for all evaluated VBDs but EHD (Fig 6). The overall risk estimate for EHD was high. Ranking of the VBDs based on the overall risk estimate is difficult given the overlapping uncertainty intervals. The results of the risk assessment were therefore also presented using a risk profile diagram (Fig 7). This diagram is a kind of P-I diagram, indicating the rate of introduction (*Probability*) on the x-axis and the economic impact of disease (*Impact*) on the y-axis as well as their uncertainty intervals. Such a risk profile diagram is very helpful to indicate the type of risk posed by the VBD. The risk profile diagram can be subdivided into four quadrants, with VBDs ending up in the upper right quadrant being of most concern since these have both a moderate to very high rate of introduction and a moderate to very high impact. Now, it can be seen that the risk of WNF is related to both an estimated high introduction rate and a high economic impact, while the risk of EHD and RVF is mainly due to an estimated high economic impact. The full uncertainty interval of WNF is in the upper right quadrant, indicating that this disease poses the highest risk to the Netherlands. The uncertainty intervals of AHS, EHD and RVF show a large overlap, making it impossible to rank these VBDs based on the results of MINTRISK, even when considering the risk profile diagram. All three diseases have huge uncertainty on the rate of

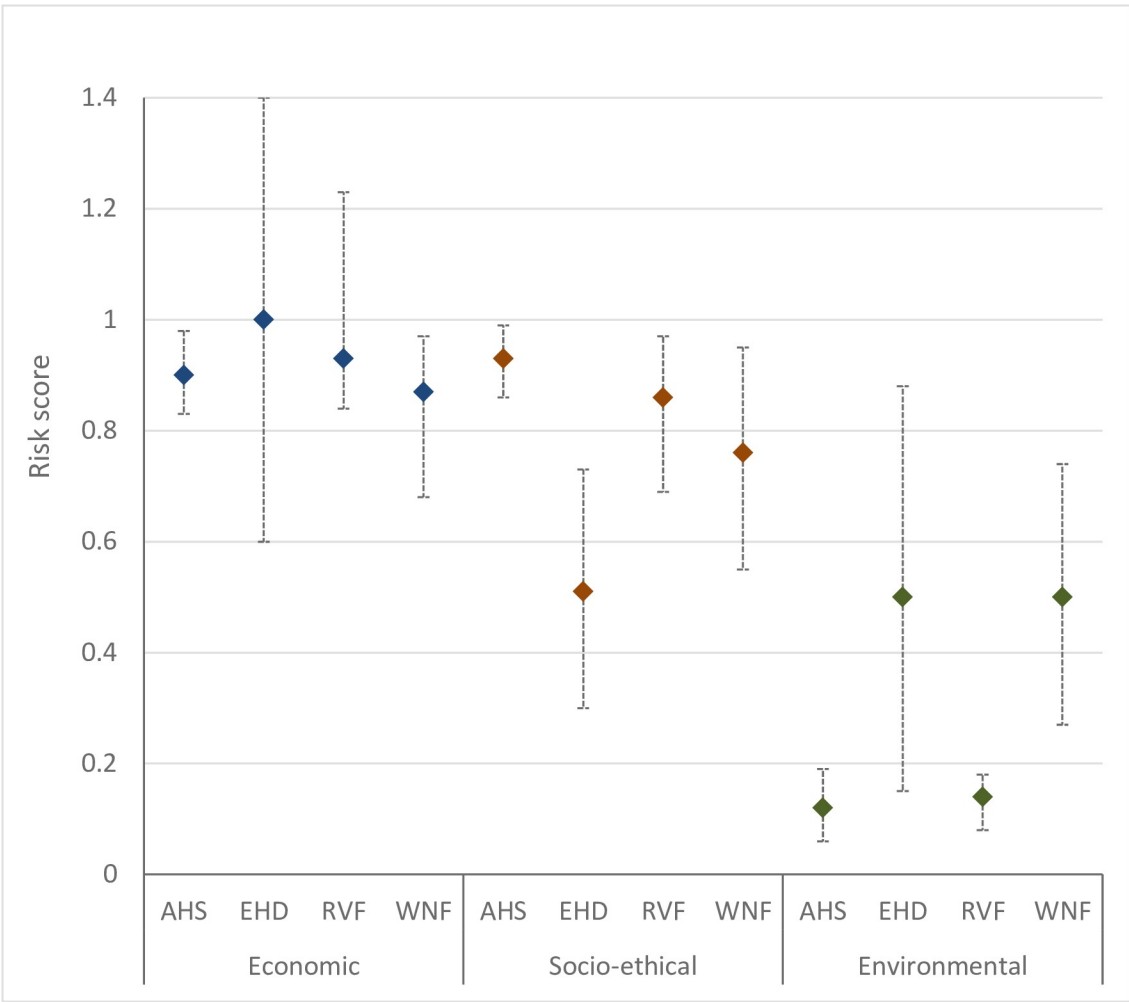

**Fig 5. Risk scores (median values) and their uncertainty (95% uncertainty interval) for the estimated impact of four vector-borne diseases for the Netherlands.** AHS = African horse sickness; EHD = epizootic haemorrhagic disease; RVF = Rift Valley fever; WNF = West Nile fever.

introduction. Uncertainty on the economic impact is, however, much higher for EHD and RVF than for AHS. This can be explained from the uncertainty on the estimated epidemic size (Fig 4). Despite the wide uncertainty intervals, it can be concluded that it is likely that economic impact is high to very high for all four diseases with the uncertainty intervals being fully located in the upper part of the diagram.

## Discussion

### Model results

Results indicate that the four VBDs considered in this study mainly differed for their rate of introduction and less for the expected economic impact of disease. Nevertheless, it is important to not only consider the probability of introduction when performing an import risk assessment, but also the impact of disease. This is even more true for VBDs if there is no competent vector or host in the area where the entry occurs or when weather conditions are not favourable for establishment. Based on the results of this study, WNF should be prioritized for

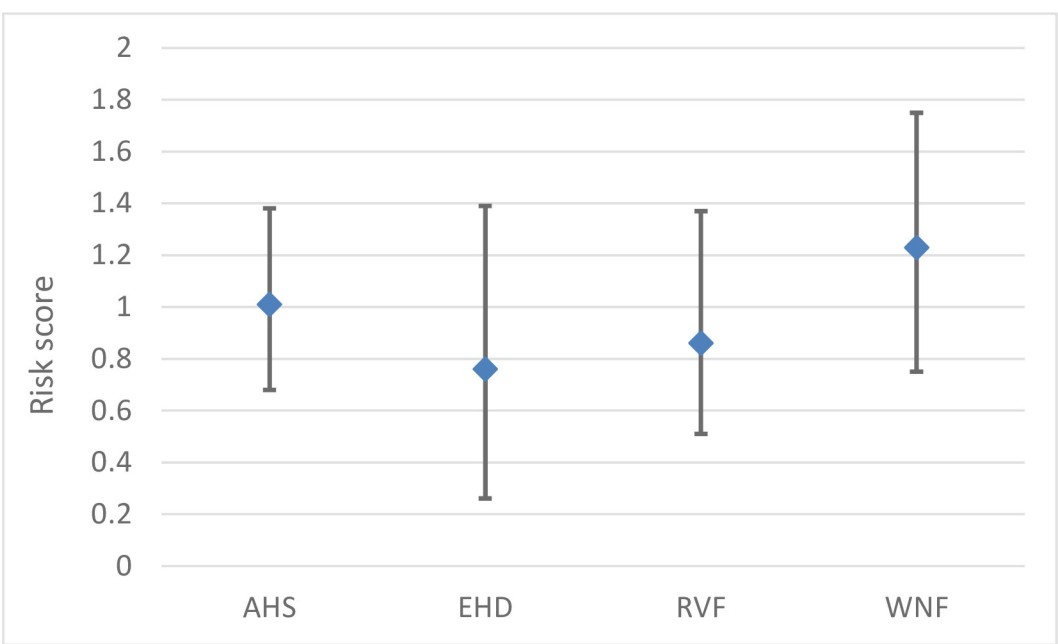

**Fig 6. Risk scores (median values) and their uncertainty (95% uncertainty interval) for the overall risk estimate of four vector-borne diseases for the Netherlands.** AHS = African horse sickness; EHD = epizootic haemorrhagic disease; RVF = Rift Valley fever; WNF = West Nile fever.

risk management in the Netherlands, both having the highest rate of introduction (Fig 3), and the highest overall risk score (Fig 6). The introduction risk of EHD was estimated to be lowest, despite this disease having the highest risk score for the economic impact (Fig 5). The relatively low rate of introduction of EHD can be explained by the fact that we limited the assessment to EHD virus serotype 6, which has a limited geographical distribution. Pathways potentially resulting in introductions from the Mediterranean, Australia and the USA were considered for EHD, with illegal importations of livestock from regions in the Mediterranean posing the highest risk. When considering the quantitative explanation of the results (S2 Appendix), an introduction of EHD or RVF resulting in establishment is expected to occur once every 100 to 1000 years, whereas a successful introduction of AHS is to be expected once every 10 to 100 years. The rate of successful WNF introductions was estimated to be even 1 to 10 times per year. It was thus not a great surprise that in August 2020, the first West Nile virus-positive bird ever was detected in the Netherlands (in a common whitethroat, *Curruca communis*) [6]. Soon after, in autumn 2020, the first autochthonous human cases of West Nile fever were reported in the Netherlands [7]. The virus had probably been circulating silently in the Dutch wild bird population for a longer period with West Nile virus-specific antibodies detected in serum samples of birds (Eurasian coot, *Fulica atra*; carrion crow, *Corvus corone*) that were collected for avian influenza surveillance in 2014–2015 [54]. This is in accordance with observations by Zehender et al. [55] that West Nile virus may be present in enzootic circulation for several years before transmission to dead-end hosts is observed.

In a risk assessment by EFSA for the European Union (EU), WNF also had a much higher rate of introduction than AHS, EHD and RVF. However, results obtained by EFSA indicated a lower introduction risk with a moderate rate of introduction for WNF and a very low rate of introduction for AHS, EHD and RVF [56]. EFSA, however, only considered imports of live animals in their assessment, whereas we also included the introduction via infected vectors or commodities. Commodities were only considered for AHS and EHD (biological products

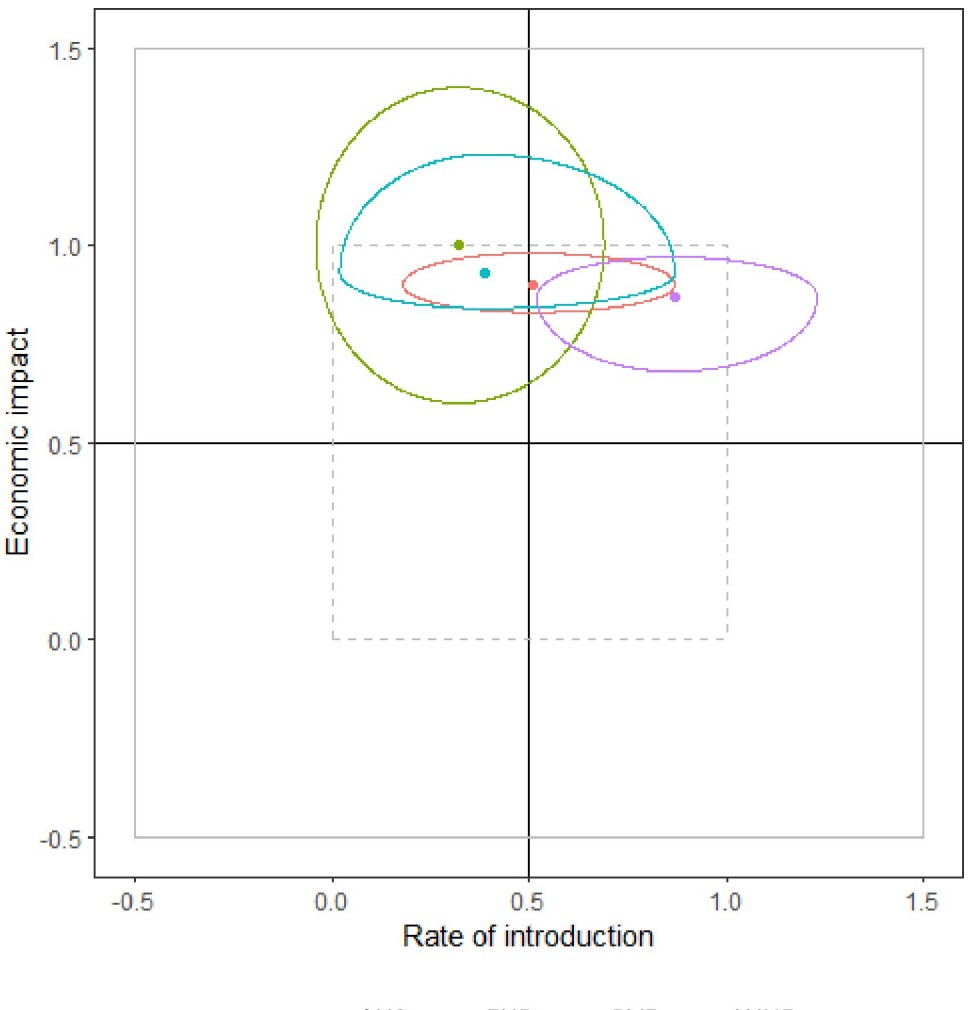

**Fig 7. Risk profile diagram indicating how the rate of introduction and the economic impact contribute to the overall risk estimate.** Dots indicate the median values for each disease with the lines enclosing the 95% uncertainty interval. Values outside the dotted square (i.e. beyond 0 and 1) indicate an extremely low (<0) or an extremely high (>1) risk. AHS = African horse sickness; EHD = epizootic haemorrhagic disease; RVF = Rift Valley fever; WNF = West Nile fever.

including modified live vaccines) and had a relatively low introduction risk and were therefore not included in the final assessment. The entry of infected vectors via various routes (aircraft, containers) appeared to be an important pathway for all VBDs evaluated in this study (Fig 3). Estimates for the numbers of vectors moved along the different pathways were, however, more uncertain than estimates of the numbers of animals imported.

The risk of RVF introduction into the EU has recently been updated by EFSA [37] given the outbreaks in the French overseas department of Mayotte [57] and the recent findings of RVF seroprevalence in Turkey [37,58,59]. They now included the introduction risk via vectors [60] and concluded that the Netherlands was among the countries having the highest rate of introduction for RVF in Europe, although the rate of introduction was still classified as low. This is similar to our result for RVF. EFSA [37] concluded that introduction of RVF is most likely via passive movement of infected vectors shipped by aircraft, containers or road transport. These pathways also had a high introduction risk in our study.

The introduction risk of VBDs for European countries was also evaluated with bespoke models. De Vos et al. [61] estimated that AHS would be successfully introduced into the Netherlands on average once every 2000 years, which is far less than the current estimate based on MINTRISK, which suggests an introduction once every 10 to 100 years. However, De Vos et al. [61] only considered legal movements of equines. In this study, the introduction risk via movement of competition horses was also evaluated, but we concluded that this pathway had negligible risk and therefore the pathway was not selected for inclusion in the final MINTRISK calculations. The current risk estimate was based on the pathways illegal import of equines and entry of midges via legal animal trade (including ruminants), pathways which were not considered by De Vos et al. [61]. In a similar study on AHS for France, Faverjon et al. [62] also concluded that the risk for legal movements of equines was low with an expected introduction every 2000 years. They, however, concluded that the risk of successful introductions via infected vectors would be even tenfold lower, which is in contrast to the results obtained with MINTRISK in this study. It should, however, be noted that Faverjon et al. [62] included an additional transmission step from vector to host in assessing the probability of establishment and considered factors such as temperature and vector abundance limiting establishment, whereas in MINTRISK we assumed favourable conditions for transmission upon entry. Brown et al. [63] estimated the probability of West Nile virus-infected mosquitoes arriving in the United Kingdom (UK) aboard aircraft from the United States (US) and concluded that this is expected to happen almost every year with on average 5 infected mosquitoes entering the UK each vector season. This estimate is in the same order of magnitude as the rate of WNF entry via this pathway into the Netherlands as estimated by MINTRISK, with a median risk score of 0.8 equalling an expected number of 10 entries annually. Note that we considered all WNF infected regions worldwide and not only the US as regions of origin of the virus. Bessell et al. [64] estimated the probability of WNF entry in Great Britain via migratory birds and concluded that this is also a high risk introduction route with an expected median value of 2 entries per year, which is in the same order of magnitude as our risk estimate for the Netherlands with a median risk score of 0.77 for the rate of entry, equalling an expected number of 7 entries per year. Rolin et al. [65] evaluated the introduction risk of RVF for the US and the EU and concluded that the most likely introduction routes for the EU would be entry via legally or illegally imported ruminants and mechanical transport of vectors in e.g. aircraft and ship cargo holds. Our assessment in MINTRISK indicated that indeed illegal imports of ruminants, and entry of mosquitoes via aircraft or containers had a relatively high risk score for the rate of introduction. However, the legal import of ruminants was assessed to have a negligible risk in this study and therefore this pathway was not selected for inclusion in the final MINTRISK calculations.

In contrast to the other VBDs, the main reservoir hosts for West Nile virus are wild birds rather than domestic animals. Evaluation of the introduction and transmission risk of WNF might therefore need additional parameters that were not included in MINTRISK. In MINTRISK, the probability of transmission and establishment is primarily based on the distribution of reservoir hosts and vectors, whereas environmental and climatic factors might be as important in evaluating these steps. Tran et al. [66], for example, identified anomalies in early summer temperatures, the presence of wetlands and location under migratory bird routes as risk factors for WNF outbreaks. Climate change might thus affect the WNF risk for The Netherlands, which is located under the Western migration flyways and has abundant wetlands. Furthermore, the evaluation of the economic impact of WNF was more difficult in MINTRISK because the agricultural losses are related to infections in equines rather than birds. The economic impact could thus not directly be linked to the estimated epidemic size, as wild birds were the epidemiological unit in the MINTRISK calculations, and not equines. Therefore, we

estimated the economic impact by scaling the expected number of equine cases to the expected number of infected birds by assuming that on average 1 infected equine host would be reported for every $10^5$ infections in wild birds and 1 infected human host for every $10^4$ infections in wild birds (number of human cases reported in 2018 ~5 times higher than the number of equine cases reported; seroprevalence in wild birds between 1 and 10%) [67–71]. These low ratios automatically resulted in selection of the 'very low' answer category for the questions addressing the economic losses per infected epidemiological unit. However, these 'very low' answer categories still result in an overestimate of the economic losses in an affected territory when expressed in Euros per infected bird. The generic approach of MINTRISK lacks flexibility to correct for this as there are only five answer categories for each question and these were scaled to account for the most likely economic losses per infected livestock animal. Another contributor to the very high estimated economic impact for WNF is the huge expected epidemic size of a WNF epidemic in the Netherlands which resulted from a relatively high transmission rate ($R_0$ between 3 and 10) and a high probability of overwintering. Estimates for transmission ($R_0$) were mainly derived from modelling studies (S3 Appendix) and these varied widely depending on e.g. ecological and climatic conditions [72,73]. Survival of an infected vector was deemed the most likely route of overwintering. However, reports on survival in mosquitoes are from areas where West Nile virus is highly prevalent [74,75] and might not be representative for the Dutch situation. As a result, the potential for spread and persistence in the Netherlands could easily have been overestimated in MINTRISK. So far, there is no indication that presence of West Nile virus in the Netherlands is resulting in extensive spread and severe economic losses, with no human or equine cases reported in 2021 up till 1 September. This could also be due to the fact that both the winter of 2020/21 and the summer of 2021 have been relatively cold compared to previous years resulting in less favourable conditions for overwintering and transmission. In general, observations from Europe show that spread of West Nile virus after recent introductions like in Germany and the Netherlands may be limited, whereas in Southern and Eastern Europe massive spread has been observed in recent years [14,76].

## Assets of MINTRISK

MINTRISK is classified as a generic risk assessment tool that can be used to assess the introduction risk for multiple diseases, allowing for prioritization of diseases for risk management [77]. In contrast to other generic risk assessment tools developed in recent years, MINTRISK was especially designed to evaluate the introduction risk of VBDs. The input and algorithms of MINTRISK put strong emphasis on the vector-host-pathogen interactions in estimating the probabilities of establishment, spread and persistence. The tool was primarily designed for livestock diseases transmitted by arthropod vectors and has been used to assess the introduction risk of diseases transmitted by midges, mosquitoes, ticks and sand flies [16,37,56, this study]. MINTRISK has, however, been most extensively tested for diseases transmitted by mosquitoes and midges and might be slightly less suited for in-depth risk assessments of tickborne diseases. Tick-borne diseases have different dynamics in transmission and persistence, with in general fewer infection generations per vector season but higher probabilities of persistence in case of transstadial transmission in the vector.

MINTRISK is one of the most complete generic risk assessment tools for disease introductions, not only addressing the probabilities of entry and exposure, but also the impact of disease over a longer period. Most generic tools in the veterinary field evaluate the introduction risk up till entry [78,79] or first infection in a new area [80–82], although some also address the epidemiological consequences [19,77,83]. MINTRISK, on the other hand, not only assesses

the epidemiological consequences, but also the economic, socio-ethical and environmental impact. This study illustrates the added value of including these impact estimates: whereas the estimated epidemic size differed widely for the four VBDs evaluated (Fig 4), the estimated economic impact was similar for all diseases (Fig 5). This is explained by the huge contribution of indirect agricultural economic losses due to, e.g., movement standstills, trade restrictions and control measures to the total economic losses induced by notifiable livestock diseases [84]. Socio-ethical and environmental impact, on the contrary, differed for the four VBDs and these estimates could also be considered by decision makers in prioritizing diseases for prevention and control.

MINTRISK is a very flexible tool that can be used for both quick and in-depth risk assessments of VBDs. For a quick assessment, the questions can be answered by experts and a first indication of the risk can be obtained as soon as all questions have been answered, regardless of the level of uncertainty entered and the number of questions answered as 'unknown'. An in-depth risk assessment, on the contrary, requires analysis of data derived from global databases on, e.g., worldwide disease occurrence and international trade, extensive literature search to estimate disease-related parameters, and expert consultation to complement any missing values. An in-depth risk assessment will in general result in narrowing of the uncertainty intervals, although uncertainty intervals can still be wide, as seen in the current assessment. This is due to the inherent nature of MINTRISK working with quantitative estimates sampled from intervals on a logarithmic scale rather than with exact values, although the tool allows the user to enter an 'own value' if an exact number is known. Even if 'low' uncertainty would be entered for all questions, i.e., input values are sampled in a range of $1 \log_{10}$ difference only, the uncertainty sampled for the individual input parameters of the model will add to a relatively high uncertainty for the overall risk estimate. The logarithmic scale used to answer questions in MINTRISK was chosen to account for the fact that values of many input parameters needed for vector-borne risk assessment are not exactly known, although the order of magnitude is often available. The logarithmic scale allows the risk assessor to provide a rather robust estimate for these parameters rather than pretending a false sense of preciseness by entering an exact value. As a consequence, results of MINTRISK are primarily useful to compare VBDs or areas at risk for their introduction risk, rather than providing an exact estimate of the introduction risk. A similar approach was used by Havelaar et al. [19] to assign boundaries to risk levels for emerging zoonotic pathogens, but they used point estimates for each level in the subsequent calculations limiting the uncertainty obtained for the resulting risk scores.

Not all questions are equally important in assessing the risk of VBDs using MINTRISK. Therefore, the risk assessor is advised to put most efforts to answer those questions that have most impact on the risk estimate. Examples would be the prevalence of disease in vectors and hosts in the risk regions; the number of vectors, hosts or commodities transported to the area at risk; the reproduction number $R_0$; and the number of infection generations in a single vector season. Although the questions on overwintering also affect the overall risk estimate, the effort on answering those questions could be limited to the most likely overwintering route as only this answer will be used to estimate the likelihood of persistence (Eq 13). MINTRISK has no built-in tool for sensitivity analysis, but the tool is very flexible to perform what-if analysis. Input values can be easily changed, and model calculations rerun. Furthermore, the attribution of the different steps in the tool to the overall risk estimate can be deduced from the risk estimates for the individual steps.

MINTRISK was developed to enable comparison of VBDs and/or areas with respect to their introduction risk in an objective, transparent and repeatable manner. This was achieved by providing quantitative explanations for the qualitative answer categories in the tool. There are, however, a few questions in MINTRISK that are very hard to quantify, and these come

without accompanying quantitative explanation. These include, e.g., questions on the socio-ethical and environmental impact of VBDs. The resulting risk scores for socio-ethical and environmental impact were therefore not used to calculate the overall risk estimate. They are, however, presented as separate output to raise awareness on possible adverse effects even if economic impact would be limited.

Pathways in MINTRISK have not been predefined to allow for flexibility accounting for the many different modes in which VBDs might be transported from risk regions to the area at risk. The risk assessor can enter multiple pathways, but only a maximum of three are used to calculate the overall risk estimate. In developing the tool, we assumed that there are only few pathways that drive the risk, especially when calculations are performed on a logarithmic scale. The risk assessor needs to select the pathways to include in the risk calculations based on the individual pathway's risk score for the rate of introduction ($RS\_Intro_{final}$). In general, the pathways with the highest value for $RS\_Intro_{final}$ will be selected. The number and type of pathways entered into MINTRISK when starting out the risk assessment is, however, decided upon by the risk assessor and pathways not entered at this stage will never show up as important, even if they would be. Hence, the initial set of pathways entered into the tool might differ between risk assessors and as such result in inconsistencies among VBD risk assessments, despite the consistency in answers given to the questions in MINTRISK. We therefore recommend to seek consensus among risk assessors on the initial set of pathways to consider. The structured questionnaire of the FEVER framework [15] provides an extensive list of possible pathways that can be used as a guidance when starting out the risk assessment. In the current risk assessment, for example, we started with a qualitative assessment of multiple pathways using FEVER and only included those with a non-negligible risk in MINTRISK.

## Conclusions

MINTRISK is a flexible tool to assess the introduction risk of VBDs in an objective, transparent and repeatable manner. The tool provides semi-quantitative risk scores that can be used for prioritization purposes. The overall risk estimate is calculated from the rate of introduction and the economic impact of disease. Results of a case study estimating the risk of four VBDs for the Netherlands indicated that the overall risk estimate was comparable for all diseases, despite the diseases having a different risk profile. Visualisation of the risk scores in a risk profile diagram allows for interpretation of these risk profiles. All diseases were estimated to have a high economic impact once introduced, but the estimated introduction rates differed, with WNF being the disease most likely to be introduced. Shortly after finishing this study, WNF was detected indeed in the Netherlands in both wild birds and humans.

## Supporting information

**S1 Appendix. Overview of input parameters in MINTRISK.**
(DOCX)

**S2 Appendix. Overview of the intermediate results for each input step and results for each output parameter of MINTRISK.**
(DOCX)

**S3 Appendix. Overview of input in MINTRISK to assess the introduction risk of four vector-borne diseases to the Netherlands.**
(DOCX)

## Acknowledgments

The authors would like to thank Barbara van der Hout (Wageningen Economic Research) for technical assistance in the development of MINTRISK.

## Author Contributions

**Conceptualization:** Clazien J. de Vos, Wil H. G. J. Hennen, Aline A. de Koeijer.

**Data curation:** Clazien J. de Vos, Herman J. W. van Roermund, Egil A. J. Fischer.

**Formal analysis:** Clazien J. de Vos.

**Funding acquisition:** Clazien J. de Vos, Sofie Dhollander, Egil A. J. Fischer, Aline A. de Koeijer.

**Methodology:** Clazien J. de Vos, Wil H. G. J. Hennen, Herman J. W. van Roermund, Sofie Dhollander, Egil A. J. Fischer, Aline A. de Koeijer.

**Project administration:** Clazien J. de Vos.

**Software:** Clazien J. de Vos, Wil H. G. J. Hennen, Aline A. de Koeijer.

**Supervision:** Clazien J. de Vos, Aline A. de Koeijer.

**Validation:** Clazien J. de Vos, Aline A. de Koeijer.

**Writing – original draft:** Clazien J. de Vos.

**Writing – review & editing:** Clazien J. de Vos, Wil H. G. J. Hennen, Herman J. W. van Roermund, Sofie Dhollander, Egil A. J. Fischer, Aline A. de Koeijer.

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
