## [Decision Letter · Decision Letter 0]

16 Aug 2021

PONE-D-21-20706

Assessing the introduction risk of vector-borne animal diseases for the Netherlands using MINTRISK: A Model for INTegrated RISK assessment

PLOS ONE

Dear Dr. de Vos,

Thank you for submitting your manuscript to PLOS ONE. After careful consideration, we feel that it has merit but does not fully meet PLOS ONE’s publication criteria as it currently stands. Therefore, we invite you to submit a revised version of the manuscript that addresses the points raised during the review process.

Both reviewers were very positive to the paper, and the amount of comments just reflects their engagement in contributing to making this a great paper for publication.

We look forward to receiving your revised manuscript.

Kind regards,

Fernanda C. Dórea

Academic Editor

PLOS ONE

Journal Requirements:

“The development of MINTRISK was funded by the Dutch Ministry of Agriculture, Nature and Food Quality (KB-12-009.01-001), Wageningen University & Research (KB-33-001-006-WBVR) and the European Food Safety Authority (NP/EFSA/ALPHA/2016/13-CT01; NP/EFSA/ALPHA/2017/10; PO/ALPHA/2019/06). The case study on vector-borne diseases was funded by the Dutch Ministry of Agriculture, Nature and Food Quality (BO-20-009-026). The authors would like to thank Barbara van der Hout (Wageningen Economic Research) for technical assistance in the development of MINTRISK.”

We note that you have provided funding information within the Acknowledgements. Please note that funding information should not appear in the Acknowledgments section or other areas of your manuscript. We will only publish funding information present in the Funding Statement section of the online submission form.

“The development of MINTRISK was funded by the Dutch Ministry of Agriculture, Nature and Food Quality (KB-12-009.01-001), Wageningen University & Research (KB-33-001-006-WBVR) and the European Food Safety Authority (NP/EFSA/ALPHA/2016/13-CT01; NP/EFSA/ALPHA/2017/10; PO/ALPHA/2019/06). The case study on vector-borne diseases was funded by the Dutch Ministry of Agriculture, Nature and Food Quality (BO-20-009-026).

URL Dutch Ministry of Agriculture, Nature and Food Quality: https://www.rijksoverheid.nl/ministeries/ministerie-van-landbouw-natuur-en-voedselkwaliteit

URL Wageningen University & Research: https://www.wur.nl/en.htm

URL European Food Safety Authority: https://www.efsa.europa.eu/en

3.Please review your reference list to ensure that it is complete and correct. If you have cited papers that have been retracted, please include the rationale for doing so in the manuscript text, or remove these references and replace them with relevant current references. Any changes to the reference list should be mentioned in the rebuttal letter that accompanies your revised manuscript. If you need to cite a retracted article, indicate the article’s retracted status in the References list and also include a citation and full reference for the retraction notice.

Reviewers' comments:

Reviewer's Responses to Questions

**Comments to the Author**

1. Is the manuscript technically sound, and do the data support the conclusions?

Reviewer #1: Yes

Reviewer #2: Yes

2. Has the statistical analysis been performed appropriately and rigorously? 

Reviewer #1: I Don't Know

Reviewer #2: N/A

3. Have the authors made all data underlying the findings in their manuscript fully available?

Reviewer #1: Yes

Reviewer #2: Yes

4. Is the manuscript presented in an intelligible fashion and written in standard English?

Reviewer #1: Yes

Reviewer #2: Yes

5. Review Comments to the Author

Reviewer #1: Over the years I have been following the development of the MINTRISK model and therefore, I read, with great interest, the manuscript, especially as it described the application of the MINTRISK model to assess the risk of introduction of specific vector borne diseases (VBDs: AHS, EHD, RVF, WNF) for a specific country (the Netherlands).

MINTRISK model includes (almost) all major parameters of the ecology and epidemiology for wide variety of VBDs transmitted by Dipteran vectors in a comprehensive way for others to appreciate; tick borne diseases fall outside its scope as described in 576-583 (since this is not a discovery of the current paper I think the authors could present MINT RISK as such from the beginning). The fact that it takes 98 specific questions to be answered in a semi-qualitative manner for each pathways to assess the risk of a specific VBD for a specific area at a certain moment is a perfect illustration of the complexity and temporal and spatial variation of VBDs. Illustrating this is for me the most important merit of the MINTRISK project and model; it forms a great help to others, just starting in the field of VBDs modeling and risk assessing, that in depth knowledge of the disease system is required to even begin to understand, assess or to project risks of VBDs in context.

The down side is that applying the model to compare risk of VBDs in your own specific situation is very labor and data intensive, while the utility of the overall risk output, besides the appreciation of the complexity is not so obvious. The authors are well aware and focus mainly on discussing the individual scores which are much more informative than overall risk output. Complements for the authors to dare to write such an elaborate manuscript and to choose not to make short cuts.

It is obvious (e.g. lines 46-47, 364, 370-371, 486, 558, 569-570) that the MINTRISK model has been developed, applied to and described for the Netherlands before West Nile virus was introduced in the Netherlands in 2020. The authors attempted to address this fact in the text (lines 495-503) but, in my opinion, this could be improved. In addition, I would to invite the authors to embrace this situation and fully address this fact and how the results are in line or what surprised them or not. This is the test case and it would be a pity to let this go to waste. I am very interested in their take on the assessment of the high economic impact (due to estimated epidemic size) of WNV in the Netherlands while there are no currently no signs for this. Maybe they can elaborate why the estimate was so high while in reality this does not seem to pan out yet.

In the following I will elaborate on more specific points in detail.

Content:

Line 36: When addressing the main result (WNV has high introduction rate) in the abstract, as a reader, I would also like to find the main factor(s) causing this in the abstract. In addition I do not think it is surprising that all four VBDs chosen for this review have high economic impact, as this was probably one of the reason why they were chosen to be included. It would have been very interesting to choose another disease and see whether the gut feeling of importance diseases for Europe also was reflected in the MintRisk tool, e.g. JEV or EEE.

Line 46-47: As stated in the general comments above it is obvious that the manuscript has been written for the larger part before the introduction of WNV in the Netherlands. Please update the manuscript (see general comment above), including adapting this sentence by adding WNF to this list of recent introduced VBDs in the Netherlands.

Line 183 & 189: The reasoning, behind the definition of an area being patchy (<5%) and homogeneous (>5%) and the accompanying value setting of Dvector, escapes me. An explanation or example of both would help me understand.

Line 240: Please replace non-susceptible animal with non-susceptible host (there are many animals in an area). In my opinion this was scored good (looking in specific in WNV)

Line 249-254: Please add a comment on the anticipated resolution of this assessment of overlap.

Line 516-518: Please add reference on the EFSA report on Assessment of the introduction RVF by vectors into Europe Van Bortel et al. 2020.

Line 541: Strange statement. Why would one only consider USA as region of origin of WNV?

Line 562-566: The reasoning is original and a interesting source for the estimate, but the spill over from the virus amplification cycle in birds-mosquitoes to horses and humans is largely determined by exposure (and sampling and reporting bias) rather than a mathematical algorithm.

Line 569-570: Could you put this outcome in perspective with the current situation in areas where WNV is introduced. There is a large difference the evolution of WNV after introductions between countries (compare Spain and France with Italy and Greece, and how does situation of Germany fit in).

Figure 1: Although it is unlikely to change, the term Rate of Introduction in the text means something else than what I would “intuitional” would think ( I would think it is synonym for entry). However the terminology is applied according the description of the paper so it is ok.

Figure 3 fig 5: Since the model is best used when comparing diseases I would put arrange the x-axes by parameter and not disease. The authors could also consider whether it is feasible and necessary to subject the data to statistical analysis whether the various risk scores are actually different between diseases.

Table 2: Why is importation of zoo-animals not considered as a source of WNV

Editorial:

Line 30: In line with the other three diseases I would refer to West Nile fever (WNF) as the disease caused by the infection with West Nile virus (and not only West Nile). Please adapt this through the manuscript.

Line 113-114: Please are write the sentence that it becomes clear that ‘very low’ and ‘very high’ are names of the categories of the answers. The current sentence is now rather confusing. In addition, make sure you unify the term; both ‘answering categories’ and ‘answer categories’ (e.g. in line 123) are used.

Line 132: Sentence is missing a word or do calculations return a risk score

Line 371: Please adjust reference numbers. Should be [11-12]. As I did not check all references, please recheck all references in the list and numbering in the text.

Table 1: I cannot find a definition of host in the text. Please add as the one of line 140 does not suffice to exclude humans, who I as a biologist define as a vertebrate animal (I do not think anthropogenic definitions helps us to understand transmission cycles).

S1-S3; The questions start with Q18. Where are Questions 1-17?

S3: I think the tables are a bit confusing as the columns of pathways are still present when they are only considered until question 51. Please adjust.

Reviewer #2: The authors present a tool called MINTRISK, which can be used to assess the risk of introduction and spread of vector-borne diseases in new areas outwith their current range. The model not only considers the potential for outbreaks in the area of concern but also the potential economic and (to a slightly lesser extent) societal impacts of vector-borne disease outbreaks if they were to occur. Applications to four diseases (African horse sickness, epizootic haemorrhagic disease Rift Valley fever and West Nile) and the risks they pose to the Netherlands are given. The work appears to be novel, extending on the previously established FEVER framework to combine different aspects of risk assessments in an objective way. There were some places where I struggled to follow how the model was constructed and so I have some suggestions for areas of clarification and/or improvement. However, whilst it may look like quite a lot of comments, I would hope that these are relatively minor and are mainly just points of clarification, as I think that this is a valuable tool which should be published.

Details of suggested amendments are given below. I have tried to score then by importance with *** denoting the most important and * denoting a fairly incidental comment in the hope that that is of help to the authors:

Introduction (Paragraph 1) (**): It seems odd to be that there is no mention that West Nile has recently been reported in the Netherlands. I notice this is mentioned in the discussion but in particular lines 46-50 don’t read right to me knowing that WN has recently been found in the Netherlands.

Material and Methods (***): There are a lot of parameter values in this paper. A table detailing what each parameter is called, what it measures, where it comes from (user input or otherwise) would be very helpful as I was constantly scrolling up and down to remember what each thing was and how it was calculated. I did realise (too late) that there is a table of questions etc. in Appendix 1 which goes some way towards doing this but what I would really like to see is a Table in the main text with all there terms involved in the Equations included.

Lines 86-88 (**): When I first read through this I was somewhat confused because it seems that transmission is simply the first half of establishment e.g. transmission is the ability of the pathogen to spread to vector to host and establishment is the ability of the pathogen to spread from vector to host and back again. This made me worry that the same thing may be modelled twice. I see now that it seems to be more of an either/or approach where the introduction risk score comes from entry*establishment if Ropt>1 but it can come from just the transmission term (Ropt) if Ropt<1. I wonder if this can be clarified earlier on to avoid my initial concerns/confusion?

Line 162 (**): “frequency of epidemics per year”?

Eqn 2 (***): This is stated to be a probability but it is not possible that it could be >1? For example, Fepi=0.5, Area=1, HRPrr=3, Prev=0.7 gives Pepi_pt=1.05. Whilst this may be an unlikely set of values I don't see why it would be implausible?

Lines 188-189 (*): The specific question in MINTRISK is "What is the estimated value of the basic reproduction ratio?", which is less descriptive than the definition here. I think it would be preferable to use this description in the MINTRISK question, as R will be affected by vector-host ratio so it may be important for users to know the assumed context?

Lines 189-191 (**): A 10% reduction in R0 when the vector is present in less than 5% of the area doesn't seem like very much - and likewise no reduction for anything more than 5% coverage doesn't seem like much. What's the justification for this choice? How sensitive are results to it?

Line 203 (*): Intuitively when I think of “introduction” I find myself thinking of what has been described in the paper as “entry”. I think it would be easier to follow if this section talked about “introduction and establishment” (although I appreciate that doesn’t fit nicely into the equations).

Lines 227-230 (**): So am I correct that each individual result is based on only a single pathway (though that pathway could be one of up to three for any given run)? Could this not lead to quite severe underestimation of risk i.e. if there are three pathways all with similar risk will that not lead to a higher overall risk (because it would be cumulative) than the case where there is one pathway with this risk and the other two have negligible risk?

Lines 254-255 (**): I don't follow this logic. Why would long-distance spread of hosts or vectors mean more infection generations in a season (or vice versa). I can see how it would increase the population at risk but not the generation time. To be clear, when I read infection generation I'm essentially thinking of the length of a gonotrophic cycle - is that the definition here? I think there needs to be a bit more justification/clarification here.

Lines 255 (**): Why 50%? How sensitive are the results to this choice?

Eqn 11 (***): This just seems to add the value for Reff in each generation but surely the number of infectives at the start of the generation will need to be included each time. For example, if Reff was 5, there were 3 generations and HRP>Tseason then this formula would give 15 infections (I think); however surely it should be 5 new infections in generation 1, then each of those 5 infections produces 5 more in generation 2 (giving 5*5=25 new infections) and then those 25 each generate a 5 new infections (giving 25*5=125 new infections). So the total number of infections would be 5+25+125=155? Unless the idea is that it is a separate introduction for each generation season and so there are 5 secondary infections each generation season with no onward spread from there but then I think the text needs to be much clearer about is meant by spread because I would interpret that is repeated introductions rather than spread. What am I missing here? Note my same confusion reappears in Eqns 12 and 14.

Lines 295-297 (**): What is the justification for focussing only on the most likely route? If there were multiple potential routes it should be possible to determine the expected number of cases of overwintering by at least one of those routes. I think what's done is fine when it's expected that one route will dominate but if there are multiple likely routes then will it not underestimate risk?

Eqn 14 (**): I think I read somewhere that you only consider the first 3 years, which would explain the 3 at the top of the last summation, however I can’t find where I read that now. It would be good to mention that next to this equation. If I didn’t read that somewhere, why only sum to 3?

Line 336 (**): Any justification for the choice of 100?

Line 346-351 (**): Why the maximum? I would have thought a cumulative measure of socio-ethical impact would be preferable in cases where one impact doesn’t dominate?

Lines 434-439 (**): Does this not suggest an issue with the model calibration if a high number of infections is able to drive a high economic impact even when the user has specified economic impact to be low? Has there been much evidence of economic impact in worse affected areas like Italy, for example?

Appendixes (**): Some explanation of the parameterisation of the log transformation would probably be helpful i.e. it is of the form a^((IV+b)*c) but it would be good to explain briefly how a, b and c are determined. Likewise for Appendix 2.

6. PLOS authors have the option to publish the peer review history of their article (what does this mean?). If published, this will include your full peer review and any attached files.

Reviewer #1: No

Reviewer #2: **Yes: **David A Ewing

---

## [Author Response · Author response to Decision Letter 0]

23 Sep 2021

We have carefully studied the guidelines and updated the manuscript’s layout to meet PLOS ONE’s style requirements.

“The development of MINTRISK was funded by the Dutch Ministry of Agriculture, Nature and Food Quality (KB-12-009.01-001), Wageningen University & Research (KB-33-001-006-WBVR) and the European Food Safety Authority (NP/EFSA/ALPHA/2016/13-CT01; NP/EFSA/ALPHA/2017/10; PO/ALPHA/2019/06). The case study on vector-borne diseases was funded by the Dutch Ministry of Agriculture, Nature and Food Quality (BO-20-009-026). The authors would like to thank Barbara van der Hout (Wageningen Economic Research) for technical assistance in the development of MINTRISK.”

We note that you have provided funding information within the Acknowledgements. Please note that funding information should not appear in the Acknowledgments section or other areas of your manuscript. We will only publish funding information present in the Funding Statement section of the online submission form.

“The development of MINTRISK was funded by the Dutch Ministry of Agriculture, Nature and Food Quality (KB-12-009.01-001), Wageningen University & Research (KB-33-001-006-WBVR) and the European Food Safety Authority (NP/EFSA/ALPHA/2016/13-CT01; NP/EFSA/ALPHA/2017/10; PO/ALPHA/2019/06). The case study on vector-borne diseases was funded by the Dutch Ministry of Agriculture, Nature and Food Quality (BO-20-009-026).

URL Dutch Ministry of Agriculture, Nature and Food Quality: https://www.rijksoverheid.nl/ministeries/ministerie-van-landbouw-natuur-en-voedselkwaliteit

URL Wageningen University & Research: https://www.wur.nl/en.htm

URL European Food Safety Authority: https://www.efsa.europa.eu/en

We have removed the information on funding from the Acknowledgements Section. The Funding Statement should read as follows:

“The development of MINTRISK was funded by the Dutch Ministry of Agriculture, Nature and Food Quality (KB-12-009.01-001), Wageningen University & Research (KB-33-001-006-WBVR) and the European Food Safety Authority (NP/EFSA/ALPHA/2016/13-CT01; NP/EFSA/ALPHA/2017/10; PO/ALPHA/2019/06). The case study on vector-borne diseases was funded by the Dutch Ministry of Agriculture, Nature and Food Quality (BO-20-009-026).”

3.Please review your reference list to ensure that it is complete and correct. If you have cited papers that have been retracted, please include the rationale for doing so in the manuscript text, or remove these references and replace them with relevant current references. Any changes to the reference list should be mentioned in the rebuttal letter that accompanies your revised manuscript. If you need to cite a retracted article, indicate the article’s retracted status in the References list and also include a citation and full reference for the retraction notice.

We reviewed the reference list and ensured that it is complete and correct.

Changes made to the reference list are:

• Two references (Sikkema et al., 2020 and Vlaskmap et al., 2020) were moved up in the reference list, because these publications are now referred to in the Introduction section. 

• One new reference (Van Bortel et al., 2020) was added to the reference list as suggested by reviewer 1. 

• Five new references (Nasci et al., 2001; Braks et al., 2017; Vogels et al., 2017; Rudolf et al., 2017; ECDC website) were added because of extending the discussion of the results for WNF.

Furthermore, a reference on the FEVER framework (De Vos et al., 2011) and two references on survival of WNV in mosquitoes during winter (Nasci et al., 2001; Rudolf et al., 2017) were added to S3 Appendix.

Response to Reviewers

We would like to thank both reviewers for carefully reading the manuscript and the useful comments they made. We appreciate the efforts they took to really understand how MINTRISK has been built. We have considered all comments made by the reviewers and revised the manuscript accordingly. 

Please note that reference is made to line numbers in the marked-up copy of the manuscript highlighting changes made to the original version (‘Revised Manuscript with Track Changes’).

Reviewer #1

Over the years I have been following the development of the MINTRISK model and therefore, I read, with great interest, the manuscript, especially as it described the application of the MINTRISK model to assess the risk of introduction of specific vector borne diseases (VBDs: AHS, EHD, RVF, WNF) for a specific country (the Netherlands).

We would like to thank this reviewer for carefully reading the manuscript and providing helpful comments, with a special focus on West Nile fever. We have considered all comments given by this reviewer and revised the manuscript accordingly.

MINTRISK model includes (almost) all major parameters of the ecology and epidemiology for wide variety of VBDs transmitted by Dipteran vectors in a comprehensive way for others to appreciate; tick borne diseases fall outside its scope as described in 576-583 (since this is not a discovery of the current paper I think the authors could present MINT RISK as such from the beginning). The fact that it takes 98 specific questions to be answered in a semi-qualitative manner for each pathways to assess the risk of a specific VBD for a specific area at a certain moment is a perfect illustration of the complexity and temporal and spatial variation of VBDs. Illustrating this is for me the most important merit of the MINTRISK project and model; it forms a great help to others, just starting in the field of VBDs modelling and risk assessing, that in depth knowledge of the disease system is required to even begin to understand, assess or to project risks of VBDs in context.

When starting MINTRISK, the scope was risk assessment of livestock diseases transmitted by arthropod vectors including ticks. Although the tool has been most extensively used (and therefore tested) for Dipteran vectors, this is not to say that the tool could not be used for tick-borne diseases at all. The tool has also been used to assess the risk of two tick-borne diseases in the past (Crimean Congo haemorrhagic fever and babesiosis) (De Vos et al., 2016). Therefore we have not presented MINTRISK as a tool for Dipteran vectors from the beginning of the manuscript.

The down side is that applying the model to compare risk of VBDs in your own specific situation is very labor and data intensive, while the utility of the overall risk output, besides the appreciation of the complexity is not so obvious. The authors are well aware and focus mainly on discussing the individual scores which are much more informative than overall risk output. Complements for the authors to dare to write such an elaborate manuscript and to choose not to make short cuts.

It is obvious (e.g. lines 46-47, 364, 370-371, 486, 558, 569-570) that the MINTRISK model has been developed, applied to and described for the Netherlands before West Nile virus was introduced in the Netherlands in 2020. The authors attempted to address this fact in the text (lines 495-503) but, in my opinion, this could be improved. In addition, I would to invite the authors to embrace this situation and fully address this fact and how the results are in line or what surprised them or not. This is the test case and it would be a pity to let this go to waste. I am very interested in their take on the assessment of the high economic impact (due to estimated epidemic size) of WNV in the Netherlands while there are no currently no signs for this. Maybe they can elaborate why the estimate was so high while in reality this does not seem to pan out yet.

In the following I will elaborate on more specific points in detail.

In the Introduction we have now added West Nile fever as one of the vector-borne diseases that were recently introduced into the Netherlands. In the Discussion section, we have elaborated on the results of MINTRISK for WNF, especially the economic impact, and tried to explain why results in MINTRISK are different from the situation observed in the Netherlands so far. As stated in the Discussion section, the assessment of the economic impact in MINTRISK is less straightforward for West Nile fever, with the reservoir hosts being wild birds rather than livestock. 

Content:

Line 36: When addressing the main result (WNV has high introduction rate) in the abstract, as a reader, I would also like to find the main factor(s) causing this in the abstract. In addition I do not think it is surprising that all four VBDs chosen for this review have high economic impact, as this was probably one of the reason why they were chosen to be included. It would have been very interesting to choose another disease and see whether the gut feeling of importance diseases for Europe also was reflected in the MintRisk tool, e.g. JEV or EEE.

We added information on the main introduction routes for WNF to the abstract. As this resulted in > 300 words, we removed the first introductory sentence of the abstract.

We agree with the reviewer that it would have been interesting to also apply MINTRISK to vector-borne diseases that are considered less of a threat for the Netherlands. The selection of vector-borne diseases was made in close cooperation with the main funder of this work, the Dutch Ministry of Agriculture, Nature and Food Quality and was based on a sense of urgency at the time the study was initiated. Adding a new disease to the current analysis is data and labour intensive indeed and was not considered essential for the current publication in which the case study is used to illustrate the application of MINTRISK.

Line 46-47: As stated in the general comments above it is obvious that the manuscript has been written for the larger part before the introduction of WNV in the Netherlands. Please update the manuscript (see general comment above), including adapting this sentence by adding WNF to this list of recent introduced VBDs in the Netherlands.

The reviewer is right that most of the manuscript was prepared before the introduction of WNV in 2020, resulting in omission of this recent incursion in the Introduction section. We have now added the introduction of West Nile in lines 48-50.

Line 183 & 189: The reasoning, behind the definition of an area being patchy (<5%) and homogeneous (>5%) and the accompanying value setting of Dvector, escapes me. An explanation or example of both would help me understand.

This reduction value was included to account for those conditions where a vector is definitely not abundantly present. However, relationships between vector abundancy and transmission are not easy to incorporate in a generic model, since transmission could still be very efficient in those areas where the vector is present indeed. If the vector is present in a few areas, the transmission will go smoothly in those few areas, while there will be very slow spatial transmission in larger areas. If R0 is sufficiently high, the spill-over from one infected sub-area will lead to spread to other subareas with vectors, but with substantial delay. If R0 is low, i.e. close to 1, there will be a lot of opportunity for fade out, even if there would be multiple epidemic starts. Thus, a limited reduction of R0 appears to be the best way to easily incorporate this aspect. This question and parameter were built into MINTRISK to raise awareness with the risk assessor that these elements have to be considered when estimating transmission of vector-borne diseases. Although Dvector will contribute to a proper estimate for the transmission rate, it cannot really account for the spatial and ecological differences within an area at risk that might either favour or hamper transmission.

We added an explanation on the parameterization of Dvector in lines 217-220.

Line 240: Please replace non-susceptible animal with non-susceptible host (there are many animals in an area). In my opinion this was scored good (looking in specific in WNV)

We changed animals into hosts. This was indeed scored correctly in MINTRISK, as we only accounted for animals that are hosts for the vectors to feed on.

Line 249-254: Please add a comment on the anticipated resolution of this assessment of overlap.

The spatial resolution to assess the overlap between vector abundance and host animal density is the area at risk considered in the risk assessment. This will often be quite a large geographical area, e.g. a country like in this study. We added “in the area at risk” to ensure that the reader is aware on which resolution the overlap is assessed.

Line 516-518: Please add reference on the EFSA report on Assessment of the introduction RVF by vectors into Europe Van Bortel et al. 2020.

We added a reference to Van Bortel et al., 2020 as suggested.

Line 541: Strange statement. Why would one only consider USA as region of origin of WNV?

This statement was made to contrast our results with those from Brown et al., 2012. They only considered commercial flights from the US, whereas we considered flights from all WN infected regions worldwide. We see no need to change our wording here.

Line 562-566: The reasoning is original and a interesting source for the estimate, but the spill over from the virus amplification cycle in birds-mosquitoes to horses and humans is largely determined by exposure (and sampling and reporting bias) rather than a mathematical algorithm.

We agree that spill over of WNV to humans or horses does not have a linear relationship with the number of infections in birds. As a result, the observed ratios that we used could be a huge overestimate (or underestimate) for the Dutch situation, partly explaining the unexpectedly high economic impact given for WNF. Since MINTRISK is a generic risk assessment tool for vector-borne diseases, that does not allow for all disease-specific details, we considered this the best way to estimate economic impact due to spill over. 

Line 569-570: Could you put this outcome in perspective with the current situation in areas where WNV is introduced. There is a large difference the evolution of WNV after introductions between countries (compare Spain and France with Italy and Greece, and how does situation of Germany fit in).

In lines 622-634 we elaborate on the assumptions for transmission (R0 value) and overwintering that we used when performing the risk assessment for WNF in MINTRISK and discussed why these might not have been correct for the Dutch situation.

Figure 1: Although it is unlikely to change, the term Rate of Introduction in the text means something else than what I would “intuitional” would think ( I would think it is synonym for entry). However the terminology is applied according the description of the paper so it is ok.

We have added clear definitions for the three summarizing output parameters to the manuscript in lines 101-106, directly after the definitions of the individual steps as we realized that these were missing in the main text. Hopefully this helps to avoid confusion with the reader.

Figure 3 fig 5: Since the model is best used when comparing diseases I would put arrange the x-axes by parameter and not disease. The authors could also consider whether it is feasible and necessary to subject the data to statistical analysis whether the various risk scores are actually different between diseases.

We agree with the reviewer that the aim of the calculations is to compare over diseases. However, for Fig 3, we did not rearrange the x-axes, since this is impossible with different pathways evaluated for each disease. For Fig 5, we appreciated the suggestion and decided to rearrange the x-axis to ease the comparison between diseases.

A statistical analysis on simulated data is usually not very helpful. In theory, one can get any difference significant if sufficient iterations are run. Therefore, we compared the diseases based on their median risk scores and their uncertainty intervals. 

Table 2: Why is importation of zoo-animals not considered as a source of WNV

Before starting the risk assessment in MINTRISK, a qualitative risk assessment was performed for each disease using the FEVER framework. FEVER provides an extensive list of potential pathways to consider including the import of exotic or zoo animals (De Vos et al., 2011). In this qualitative assessment we did consider importation of zoo animals as a source of WNV, but concluded that the probability of entry and establishment was very low. Two important differences with migratory birds are the low numbers involved and the fact that zoo birds are subjected to import regulations. Pathways with a very low probability for introduction in the qualitative assessment were not selected for the semi-quantitative assessment in MINTRISK.

Editorial:

Line 30: In line with the other three diseases I would refer to West Nile fever (WNF) as the disease caused by the infection with West Nile virus (and not only West Nile). Please adapt this through the manuscript.

Thanks for this useful suggestion. We have changed West Nile (WN) into West Nile fever (WNF) throughout the manuscript.

Line 113-114: Please are write the sentence that it becomes clear that ‘very low’ and ‘very high’ are names of the categories of the answers. The current sentence is now rather confusing. In addition, make sure you unify the term; both ‘answering categories’ and ‘answer categories’ (e.g. in line 123) are used.

We changed the order of the sentence in line 113-114 (now lines 129-131) and put the answer categories between quotes to avoid any confusion for the reader.

We thank the reviewer for spotting inconsistencies in our wording when indicating the answer categories. We changed answering categories into answer categories throughout the manuscript.

Line 132: Sentence is missing a word or do calculations return a risk score

Calculations do indeed return a semi-quantitative risk score for each individual step in MINTRISK as described in lines 142-144. We added semi-quantitative to line 132 (now line 149) and hope it is clearer now. 

Line 371: Please adjust reference numbers. Should be [11-12]. As I did not check all references, please recheck all references in the list and numbering in the text.

Thanks for noticing. We adjusted the reference numbers here and checked all references and their numbering throughout the manuscript. No other mistakes were found. Please note that some reference numbers have been updated in the revised manuscript after inclusion of new references. 

Table 1: I cannot find a definition of host in the text. Please add as the one of line 140 does not suffice to exclude humans, who I as a biologist define as a vertebrate animal (I do not think anthropogenic definitions helps us to understand transmission cycles).

We agree with the reviewer that vertebrate hosts would include humans if susceptible. We changed the caption and headings in Table 1 to make clear that we only indicate vertebrate host animals in the third column. Whether or not humans are a vertebrate host is indicated in the fifth column (zoonosis). We included a comment in the first paragraph of the Material and Methods (lines 87-89) to make explicit that MINTRISK models the infection dynamics between arthropod vectors and vertebrate host animals, and not humans.

S1-S3; The questions start with Q18. Where are Questions 1-17?

Numbering of questions in MINTRISK was aligned with numbering of questions in FEVER. The FEVER framework starts with a hazard identification, based on 17 questions. Since the questions of the hazard identification are not used to semi-quantitatively estimate the introduction risk, those questions are missing from MINTRISK. We added a footnote on this to S1 and S3 Appendices.

S3: I think the tables are a bit confusing as the columns of pathways are still present when they are only considered until question 51. Please adjust.

We have now split the tables into two: one for the first three steps in MINTRISK (entry, transmission and establishment) in which questions are answered by pathway, and one for the second three steps (spread, persistence and impact) in which answers to the questions are independent of pathways.

Reviewer #2 

The authors present a tool called MINTRISK, which can be used to assess the risk of introduction and spread of vector-borne diseases in new areas outwith their current range. The model not only considers the potential for outbreaks in the area of concern but also the potential economic and (to a slightly lesser extent) societal impacts of vector-borne disease outbreaks if they were to occur. Applications to four diseases (African horse sickness, epizootic haemorrhagic disease Rift Valley fever and West Nile) and the risks they pose to the Netherlands are given. The work appears to be novel, extending on the previously established FEVER framework to combine different aspects of risk assessments in an objective way. There were some places where I struggled to follow how the model was constructed and so I have some suggestions for areas of clarification and/or improvement. However, whilst it may look like quite a lot of comments, I would hope that these are relatively minor and are mainly just points of clarification, as I think that this is a valuable tool which should be published.

We would like to thank this reviewer for carefully reading the manuscript and providing helpful comments. We appreciate the efforts made by this reviewer to fully understand the mathematical equations of MINTRISK. We have considered all comments made by the reviewer and revised the manuscript accordingly. 

Details of suggested amendments are given below. I have tried to score then by importance with *** denoting the most important and * denoting a fairly incidental comment in the hope that that is of help to the authors:

Introduction (Paragraph 1) (**): It seems odd to be that there is no mention that West Nile has recently been reported in the Netherlands. I notice this is mentioned in the discussion but in particular lines 46-50 don’t read right to me knowing that WN has recently been found in the Netherlands.

The reviewer is right that the recent introduction of West Nile in the Netherlands should also be stated in the Introduction section. We have added the introduction of West Nile in lines 48-50.

Material and Methods (***): There are a lot of parameter values in this paper. A table detailing what each parameter is called, what it measures, where it comes from (user input or otherwise) would be very helpful as I was constantly scrolling up and down to remember what each thing was and how it was calculated. I did realise (too late) that there is a table of questions etc. in Appendix 1 which goes some way towards doing this but what I would really like to see is a Table in the main text with all there terms involved in the Equations included.

We compiled a table with an overview of parameters in MINTRISK as suggested by the reviewer and added this table (Table 1) to the main body of the manuscript.

Lines 86-88 (**): When I first read through this I was somewhat confused because it seems that transmission is simply the first half of establishment e.g. transmission is the ability of the pathogen to spread to vector to host and establishment is the ability of the pathogen to spread from vector to host and back again. This made me worry that the same thing may be modelled twice. I see now that it seems to be more of an either/or approach where the introduction risk score comes from entry*establishment if Ropt>1 but it can come from just the transmission term (Ropt) if Ropt<1. I wonder if this can be clarified earlier on to avoid my initial concerns/confusion?

MINTRISK is based on the FEVER framework. In that framework, we advised risk assessors to first estimate the steps for entry and transmission and to only proceed with the risk assessment if both of them were non-negligible. That’s why transmission is estimated in an early stage in MINTRISK as well. We have added a sentence explaining this rationale to the manuscript in lines 204-205. Furthermore, the reviewer’s comment made us realise that the position of the transmission and establishment steps were not correctly positioned in Fig. 1 and therefore we slightly amended Fig 1. 

Line 162 (**): “frequency of epidemics per year”?

Yes indeed. We added “per year” to this line.

Eqn 2 (***): This is stated to be a probability but it is not possible that it could be >1? For example, Fepi=0.5, Area=1, HRPrr=3, Prev=0.7 gives Pepi_pt=1.05. Whilst this may be an unlikely set of values I don't see why it would be implausible?

The reviewer is right that, in theory, the value of Pepi_pt could be > 1. Whereas the values of Area and Prevepi_pt are always < 1, the values of Fepi and HRPRR are not. However, the latter two values are dependent on each other. If the frequency of epidemics would be > 1, then the length of the high risk period is by definition < 1 and vice versa. (The example above is thus not a realistic set of parameter values, suggesting one epidemic every 2 years, but detection of each epidemic only after a period of 3 years) Hence, a sensible set of input parameters will not result in a probability > 1 for Pepi_pt. Furthermore, values for Prevepi_pt are in general very low, even further reducing the possibilities for Pepi_pt being > 1. To ensure that the value of Pepi_pt does not exceed 1, we changed the equation in both the manuscript and the model code into: 

P_(epi_pt)=Min(F_epi×Area×〖HRP〗_RR×〖Prev〗_(epi_pt),1)

Lines 188-189 (*): The specific question in MINTRISK is "What is the estimated value of the basic reproduction ratio?", which is less descriptive than the definition here. I think it would be preferable to use this description in the MINTRISK question, as R will be affected by vector-host ratio so it may be important for users to know the assumed context?

This has been accounted for in MINTRISK by providing additional information under the i-button. We added an additional comment in MINTRISK that R0 should be estimated for a fully susceptible host population. Users of MINTRISK are encouraged to read this additional information before answering each question.

Lines 189-191 (**): A 10% reduction in R0 when the vector is present in less than 5% of the area doesn't seem like very much - and likewise no reduction for anything more than 5% coverage doesn't seem like much. What's the justification for this choice? How sensitive are results to it?

The reviewer is right in that this will not result in a high reduction of R0 and hence will not severely affect MINTRISK’s estimate for transmission. This reduction value was included to account for those conditions where a vector is definitely not abundantly present. However, relationships between vector abundancy and transmission are not easy to incorporate in a generic model, since transmission might still be very efficient in those areas where the vector is present indeed. If the vector is present in a few areas, the transmission will go smoothly in those few areas, while there will be very slow spatial transmission in larger areas. If R0 is sufficiently high, the spill-over from one infected sub-area will lead to spread to other subareas with vectors, but with substantial delay. If R0 is low, i.e. close to 1, there will be a lot of opportunity for fade out, even if there would be multiple epidemic starts. Thus, a limited reduction of R0 appears to be the best way to easily incorporate this aspect. This question and parameter were built into MINTRISK to raise awareness with the risk assessor that these elements have to be considered when estimating transmission of vector-borne diseases. Although Dvector will contribute to a proper estimate for the transmission rate, it cannot account for the spatial and ecological differences within an area at risk that might either favour or hamper transmission.

We added an explanation on the parameterization of Dvector in lines 217-220.

Line 203 (*): Intuitively when I think of “introduction” I find myself thinking of what has been described in the paper as “entry”. I think it would be easier to follow if this section talked about “introduction and establishment” (although I appreciate that doesn’t fit nicely into the equations).

Thanks for noting this. We agree that entry and introduction are very close (maybe even synonyms) and that using introduction for the successful entry of a vector-borne disease (i.e. entry and establishment) might be somewhat confusing. We have tried to avoid confusion by giving clear definitions (lines 91-100) and providing Fig 1. However, upon rereading we realised that it might be helpful for the reader to include definitions for the summarizing output parameters as well at the start of the model description. These were therefore added to the manuscript in lines 101-106, directly after the definitions of the individual steps.

Lines 227-230 (**): So am I correct that each individual result is based on only a single pathway (though that pathway could be one of up to three for any given run)? Could this not lead to quite severe underestimation of risk i.e. if there are three pathways all with similar risk will that not lead to a higher overall risk (because it would be cumulative) than the case where there is one pathway with this risk and the other two have negligible risk?

The reviewer is correct that results of each iteration are based on a single pathway only, and that this pathway can vary among the (max 3) pathways selected. Since model input and output is given on a logarithmic scale, the pathway with the highest rate of introduction will in general drive the overall risk estimate, even if results of the individual pathways were added. The reviewer is right that the risk would be slightly higher if all three pathways had comparable risk (which is not a very common scenario), but even then, it would be at most 0.5 times higher than the calculated risk based on a single pathway. 

When revising the paper, we realised that the logarithmic scale of answer categories and output was not communicated very clearly in the manuscript (although given in S1 and S2 Appendices). We therefore added a comment on this in lines 116-118.

Lines 254-255 (**): I don't follow this logic. Why would long-distance spread of hosts or vectors mean more infection generations in a season (or vice versa). I can see how it would increase the population at risk but not the generation time. To be clear, when I read infection generation I'm essentially thinking of the length of a gonotrophic cycle - is that the definition here? I think there needs to be a bit more justification/clarification here.

These parameters (Overlap, Local, Movvector and Movhost) affect the rate at with which the infection will spread in the area at risk. The rate of transmission is both determined by the R0 value and the number of infection generations in a vector season. We have chosen to model the effect of these parameters on disease spread by reducing the number of infection generations in case these parameters are limiting the efficient transmission of the infection. Less infection generations (and thus a longer time span between infection generations) will reduce the rate of transmission in the vector season and therefore also the number of infected host animals (epidemiological units). We have added a brief explanation on how these parameters would affect the number of infection generations in lines 282-289.

An infection generation can be compared to a generation in population biology, based on R0. Each next generation of infections is a new infection generation. The time needed from a “parent generation” to its “offspring generation” depends on the latent and infectious period in host animals and the extrinsic incubation period and infectious period (� lifespan) in vectors. The number of infection generations in one vector season is based on these parameters and the estimated length of the vector season. We added this definition to the information on this question in MINTRISK under the i-button to ensure that the risk assessor has a good understanding of infection generations.

Lines 255 (**): Why 50%? How sensitive are the results to this choice?

This 50% is an arbitrary choice. We deemed it unrealistic to include a larger reduction effect. All those aspects lead to slower transmission, which we incorporate into fewer infection generations per year, but they cannot stop transmission as such, since there are many routes of transmission possible in these vector borne diseases. In most assessments, these four parameters (Overlap, Local, Movvector and Movhost) will only result in a slight reduction of the number of infection generations.

Eqn 11 (***): This just seems to add the value for Reff in each generation but surely the number of infectives at the start of the generation will need to be included each time. For example, if Reff was 5, there were 3 generations and HRP>Tseason then this formula would give 15 infections (I think); however surely it should be 5 new infections in generation 1, then each of those 5 infections produces 5 more in generation 2 (giving 5*5=25 new infections) and then those 25 each generate a 5 new infections (giving 25*5=125 new infections). So the total number of infections would be 5+25+125=155? Unless the idea is that it is a separate introduction for each generation season and so there are 5 secondary infections each generation season with no onward spread from there but then I think the text needs to be much clearer about is meant by spread because I would interpret that is repeated introductions rather than spread. What am I missing here? Note my same confusion reappears in Eqns 12 and 14.

Equations 11, 12 and 14 are pretty complex equations indeed. We will try to explain using [Disp-formula pone.0259466.e011] as an example. In the first line, i.e. if the HRP is longer than the vector season, the formula results in the sum of the number of infected animals in each new infection generation, which would be 1 to start with, Reff in the second generation, Reff^2 in the third generation etc. So, if Reff was 5 and IGeff was 3, this would result in a total of 1 + 5 + 25 +125 infections indeed. In the second line, i.e. if the HRP is shorter than the length of the vector season, the number of infected epidemiological units is calculated in two parts. First for the period before detection. This is equal to the calculation in the first line, but now only for the infection generations until detection of the disease (IGdet). Then the second part starts with the number of infected animals in the last generation before detection, which equals Reff^IGdet and this is multiplied with the expected number of animals in the next generations until the end of the vector season when control measures are in place. This number is calculated similarly to the number of infected animals before detection by summing the number of infected animals in each new infection generation. 

In explaining [Disp-formula pone.0259466.e011] to the reviewer, we realised that the summations should start with i=0 and not i=1. We changed this in the equation and checked the model code. In the model code, this was programmed correctly, so there was no need to update the calculations for the four vector-borne diseases.

Lines 295-297 (**): What is the justification for focussing only on the most likely route? If there were multiple potential routes it should be possible to determine the expected number of cases of overwintering by at least one of those routes. I think what's done is fine when it's expected that one route will dominate but if there are multiple likely routes then will it not underestimate risk?

Analogue to selecting only the pathway with the highest introduction score (see the reviewer’s comment for lines 227-230), we here also select only the overwintering route with the highest probability, because of the logarithmic scale at which input and output of MINTRISK is given. 

Eqn 14 (**): I think I read somewhere that you only consider the first 3 years, which would explain the 3 at the top of the last summation, however I can’t find where I read that now. It would be good to mention that next to this equation. If I didn’t read that somewhere, why only sum to 3?

In the text below [Disp-formula pone.0259466.e014], we state that the epidemic size is calculated over a total of four vector seasons. This explains the 3 at the top of the last summation, as the first year is indicated by i=0. We have now included definitions for the summarizing output parameters in lines 101-106 (in response to a comment of reviewer 1), where it is also stated that the epidemic size is calculated over four vector seasons.

Line 336 (**): Any justification for the choice of 100?

To calculate the economic impact of a zoonotic vector-borne disease, the economic costs due to human disease had to be offset to the number of infected animal hosts. Since humans are mostly spill over hosts, resulting in relatively few cases, we deemed it more easy to estimate the costs per 100 infected animals. The number of 100 was a bit of an arbitrary choice. It worked pretty well for RVF, since the number of expected human cases was estimated to be 1 for every 100 infected animals. For West Nile fever, the ratio between cases in humans and infections in birds is, however, lower with 1 expected human case for every 10^4 - 10^5 infected birds.

Line 346-351 (**): Why the maximum? I would have thought a cumulative measure of socio-ethical impact would be preferable in cases where one impact doesn’t dominate?

Analogue to selecting only the pathway with the highest introduction score (see the reviewer’s comment for lines 227-230) and to selecting only the overwintering route with the highest probability (see the reviewer’s comment for lines 295-297), we here also selected only the socio-ethical (and environmental) impact with the highest risk score, because of the logarithmic scale at which input and output of MINTRISK is given. 

Lines 434-439 (**): Does this not suggest an issue with the model calibration if a high number of infections is able to drive a high economic impact even when the user has specified economic impact to be low? Has there been much evidence of economic impact in worse affected areas like Italy, for example?

Calculation of the economic impact in MINTRISK is partly based on the multiplication of the epidemic size with the costs per epidemiological unit ([Disp-formula pone.0259466.e015]). For the case of West Nile fever, the epidemiological units are wild birds, and not horses. Therefore we had to estimate the economic loss per infected wild bird, which will be very low indeed. The reviewer is right that we here encounter a potential issue with model scaling (not calibration though), since the five answer categories for economic losses per epidemiological unit were scaled to account for losses per infected livestock animal. As a consequence, even the very low answer category will probably have overestimated the economic loss per epidemiological unit (bird) for WNF. The ‘very low’ answer category still results in a loss between 1 and 10 Euros per infected bird, and this is repeated for three parameters (EcoDA, EcoIA, EcoPH). Using lower values for the answer categories of these input parameters would, however, hamper a proper assessment for diseases for which livestock animals are the reservoir host and thus the epidemiological unit in MINTRISK, since economic losses per animal can be high indeed, like for AHS. We conclude that the economic assessment for West Nile fever is providing less reliable results because of this issue. We slightly changed the wording in lines 474-477 to make clear that wild birds are the epidemiological units for the assessment and the issue with model scaling is further discussed in the Discussion section in lines 606-619.

Appendixes (**): Some explanation of the parameterisation of the log transformation would probably be helpful i.e. it is of the form a^((IV+b)*c) but it would be good to explain briefly how a, b and c are determined. Likewise for Appendix 2.

An explanation of the general equations used for the log-transformation and inverse log-transformation were added to footnote c in S1 Appendix and footnote a in S2 Appendix, respectively.

---

## [Decision Letter · Decision Letter 1]

20 Oct 2021

Assessing the introduction risk of vector-borne animal diseases for the Netherlands using MINTRISK: A Model for INTegrated RISK assessment

PONE-D-21-20706R1

Dear Dr. de Vos,

We’re pleased to inform you that your manuscript has been judged scientifically suitable for publication and will be formally accepted for publication once it meets all outstanding technical requirements.

Please check the one small last issue pointed out by reviewer 2, and if it needs correction, address it during production. 

Kind regards,

Fernanda C. Dórea

Academic Editor

PLOS ONE

Additional Editor Comments (optional):

Reviewers' comments:

Reviewer's Responses to Questions

**Comments to the Author**

1. If the authors have adequately addressed your comments raised in a previous round of review and you feel that this manuscript is now acceptable for publication, you may indicate that here to bypass the “Comments to the Author” section, enter your conflict of interest statement in the “Confidential to Editor” section, and submit your "Accept" recommendation.

Reviewer #1: All comments have been addressed

Reviewer #2: All comments have been addressed

2. Is the manuscript technically sound, and do the data support the conclusions?

Reviewer #1: Yes

Reviewer #2: Yes

3. Has the statistical analysis been performed appropriately and rigorously? 

Reviewer #1: Yes

Reviewer #2: Yes

4. Have the authors made all data underlying the findings in their manuscript fully available?

Reviewer #1: (No Response)

Reviewer #2: Yes

5. Is the manuscript presented in an intelligible fashion and written in standard English?

Reviewer #1: Yes

Reviewer #2: Yes

6. Review Comments to the Author

Reviewer #1: The authors took the points of both reviewers to heart and addressed them accordingly.

One small but important issue remains: Line 34 -36 Are the authors sure that this statements about likely route of virus by mosquitoes in an aircraft concerns West Nile virus and not Rift valley fever virus?

Reviewer #2: (No Response)

7. PLOS authors have the option to publish the peer review history of their article (what does this mean?). If published, this will include your full peer review and any attached files.

Reviewer #1: **Yes: **Marieta A.H. Braks

Reviewer #2: **Yes: **David Ewing

---

## [Editor Report · Acceptance letter]

25 Oct 2021

PONE-D-21-20706R1 

Assessing the introduction risk of vector-borne animal diseases for the Netherlands using MINTRISK: A Model for INTegrated RISK assessment 

Dear Dr. de Vos:

I'm pleased to inform you that your manuscript has been deemed suitable for publication in PLOS ONE. Congratulations! Your manuscript is now with our production department. 

Kind regards, 

on behalf of

Dr. Fernanda C. Dórea 

Academic Editor

PLOS ONE